# MoSH: Modeling Multi-Objective Tradeoffs with Soft and Hard Bounds

## Abstract

Countless science and engineering applications in multi-objective optimization (MOO) necessitate that decision-makers (DMs) select a Pareto-optimal solution which aligns with their preferences. Evaluating individual solutions is often expensive, necessitating cost-sensitive optimization techniques. Due to competing objectives, the space of trade-offs is also expansive — thus, examining the full Pareto frontier may prove overwhelming to a DM. Such real-world settings generally have loosely-defined and context-specific desirable regions for each objective function that can aid in constraining the search over the Pareto frontier. We introduce a novel conceptual framework that operationalizes these priors using *soft-hard functions*, SHFs, which allow for the DM to intuitively impose soft and hard bounds on each objective – which has been lacking in previous MOO frameworks. Leveraging a novel minimax formulation for Pareto frontier sampling, we propose a two-step process for obtaining a compact set of Pareto-optimal points which respect the user-defined soft and hard bounds: (1) densely sample the Pareto frontier using Bayesian optimization, and (2) sparsify the selected set to surface to the user, using robust submodular function optimization. We prove that (2) obtains the optimal compact Pareto-optimal set of points from (1). We further show that many practical problems fit within the SHF framework and provide extensive empirical validation on diverse domains, including brachytherapy, engineering design, and large language model personalization. Specifically, for brachytherapy, our approach returns a compact set of points with over 3% greater SHF-defined utility than the next best approach. Among the other diverse experiments, our approach consistently leads in utility, allowing the DM to reach >99% of their maximum possible desired utility within validation of 5 points.

## 1 Introduction

Various critical real-world applications of optimization, including healthcare, drug discovery, engineering design, and deep learning, involve optimizing over multiple, often expensive, and competing objectives $f_1(x),...,f_L(x)$, termed multi-objective optimization (MOO) (Fromer & Coley, 2023; Luukkonen et al., 2023; Xie et al., 2021; Yu et al., 2000; Papadimitriou & Yannakakis, 2001). In general, the intention in such real-world applications is to select a single set of usable parameters that lie on the Pareto frontier (PF) – parameters that lead to the ideal set of trade-offs as determined by some decision-maker (DM). However, due to the often continuous and competing nature of multiple objectives, searching over the entire space of trade-offs is unmanageable. Thus, selecting the ideal point along the Pareto frontier takes the form of an iterative process by which the DM explores possible trade-offs prior to making an informed final decision which satisfies their preferences (Liu et al., 2021).

In the healthcare domain, MOO problems commonly appear as the interplay between maximizing targeted treatment while limiting harmful side-effects in a patient. Electing a suitable set of trade-offs that correspond to dominant clinical opinion is therefore crucial, as the result is high-risk and significantly impactful to patients. The use of brachytherapy, internal radiation therapy for cancer treatment, presents an example of this kind (Deufel et al., 2020). In brachytherapy, the clinician must find the optimal treatment plan that balances maximal radiation dosage level in the cancer tumor with minimal damage to surrounding healthy organs. The time-consuming nature of devising each individual treatment plan and the immensity of the trade-off space in radiation dosage levels

typically results in the clinician spending much of their valuable time exploring that patient-specific space, before deciding on a treatment plan which suits their clinical preferences (Bélanger et al., 2019; Cui et al., 2018; van der Meer et al., 2020). Oftentimes, the clinician has a rough idea of the desired dosage levels for the treatment plan, e.g. cover at least 90% of the tumor but ideally over 95%, and preferably emit less than 513 centigrays (cGY) of radiation to the bladder, with a strict upper limit of 601 cGY (Viswanathan et al., 2012). In these time-critical settings, treatment planning would rather involve providing the clinician with a small, and thus easily navigable, set of treatment plans that respect such constraints. Despite the significant progress in MOO, methods which employ this intuitive notion of multi-level constraints for obtaining compact Pareto-optimal (PO) sets remain underexplored (Paria et al., 2019; Abdolshah et al., 2019; Suzuki et al., 2020; Zuluaga et al., 2016; While et al., 2012). We use the application of brachytherapy to motivate our MOO framework, and leverage real patient data and expert clinical knowledge in a subset of our experiments to demonstrate the practical utility of our introduced method.

In this paper, we present a novel conceptual framework that allows for the DM to exploit the intuitive and prevalent notion of soft and hard preferences to obtain an easily traversable set of Pareto-optimal (PO) points, using utility functions which we term *soft-hard functions*, or SHFs. We further assume the DM maintains a set of (unknown) trade-off preferences, parameterized by some prior distribution over the PF constrained implicitly by SHFs. In particular, we assume a minimum desirable level for each $f_\ell(x)$, represented by hard bounds, and a diminishment threshold for each $f_\ell(x)$, above which the DM becomes relatively indifferent to increases, represented by soft bounds. As verification of each PO point is costly, we introduce a novel formulation for maximizing such utilities on the PF and propose a two-step process for obtaining a small, compact set of PO points within the afore-mentioned setting: (1) densely sample the PF using Bayesian optimization, and (2) sparsify the set of points from (1) for presentation to the DM, using robust submodular function optimization. We further propose a set of soft-hard metrics and empirically and theoretically validate both (1) and (2); namely, we show that step (2) theoretically guarantees to obtain the near optimal set of points, from (1), which is robust to the worst-case value of the DM's unknown preferences.

Although there exists an extensive line of work on multi-objective optimization, most works focus on populating the entire PF (Ponweiser et al., 2008; Hernández-Lobato et al., 2016). Several recent works attempt to recover a subset of the PF by assuming different forms of priors or thresholds, however, they lack the same capabilities which our method affords for the DM – a highly intuitive notion of soft and hard bounds, for each of the objectives, which return an easily navigable set of Pareto optimal points (Paria et al., 2019; Malkomes et al., 2021). In summary, our contributions comprise the following:

1. We introduce the **novel conceptual framework of soft-hard bounds** to capture the ubiq-uitous but previously unmodeled notion that practitioners typically have both hard and soft requirements for each objective. We formulate this as a novel minimax optimization prob-lem which aims to maximize the SHFs of the DM under an unknown set of preferences.

2. We introduce a **two-step process to solve our proposed formulation and obtain a small, and diverse set of Pareto-optimal points which reflect the soft and hard bounds**: (1) use multi-objective Bayesian optimization to obtain a, theoretically guaranteed, set of densely sampled points on the PF, and (2) use robust submodular function optimization to sparsify the points from (1) while maintaining theoretical guarantees on the near optimality of the sparse set of points, with respect to the DM's unknown preferences and SHFs.

3. We conduct **extensive empirical evaluations, using our proposed soft-hard metrics, on a novel set of synthetic and real-world applications, spanning engineering design, large language model (LLM) personalization, deep learning model selection, and a real clinical case for cervical cancer brachytherapy treatment planning**. Specifically, in cervical cancer brachytherapy, we show that our approach returns a compact set of treat-ment plans which offers over 3% greater SHF-defined utility than the next best approach. Among the other diverse experiments, our approach also consistently achieves the leading utility, allowing the DM to reach >**99% of their maximum possible desired utility** within validation of 5 points.

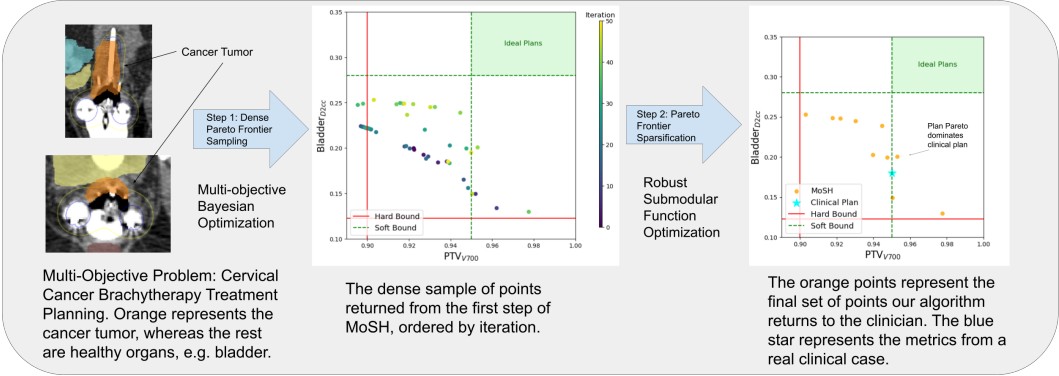

Figure 1: Overall pipeline for our proposed method, MoSH. We evaluated MoSH on a real clinical case for cervical cancer brachytherapy treatment planning, where the objectives are to balance between the radiation dosage levels to the cancer tumor and to the nearby healthy organs – bladder, rectum, and bowel. For the plots, we only showed two of the four dimensions in the multi-criterion objective for this task. In step 1, the iterations convey the gradual manner in which the sampled points move towards the clinician's region of ideal plans. In step 2, the Pareto dominant plan's metrics surpass those of the plan from an expert clinician in all dimensions, on real data.

## 2 MULTI-OBJECTIVE OPTIMIZATION WITH SOFT-HARD FUNCTIONS

In this section, we outline the primary optimization setting we consider and introduce notation used throughout the paper in §2.1. We develop the intuition and motivation for SHFs and define them explicitly in §2.2. We thus arrive at the problem definition through our mini-max formulation in §2.3.

### 2.1 MULTI-OBJECTIVE OPTIMIZATION BACKGROUND

As the name suggests, a multi-objective optimization (MOO) problem is an optimization problem that concerns multiple objective functions. Any MOO problem can be written as the joint maximization of $L$ objective functions over some input space $X \subset \mathbb{R}^d$,

$$\max_{x \in X}(f_1(x), \dots, f_L(x))$$

in which each $f_\ell$, $\ell \in [L]$, defines a function $f_\ell : X \to \mathbb{R}$. Broadly speaking, there does not typically exist a *feasible* solution that marginally optimizes each objective function simultaneously. Therefore, work in MOO generally focuses on Pareto-optimal (PO) solutions. A feasible solution $x^\dagger$ is considered *Pareto-optimal* if no objective can improve without degrading another; in other words, if $x^\dagger$ is not *Pareto-dominated* by any other solution (see definition in Appendix A.3).

A common approach to multi-objective optimization is to convert the $L$-dimensional objective to a scalar in order to utilize standard optimization methods via a scalarization function. Scalarization functions typically take the form $s_\lambda : \mathbb{R}^L \to \mathbb{R}$, parameterized by $\boldsymbol{\lambda}$ from some set $\Lambda$ in $L$-dimensional space (Roijers et al., 2013; Paria et al., 2019). For instance, the general class of linear scalarization functions $s_\Lambda(\boldsymbol{y}) := \{\boldsymbol{\lambda}^\top \boldsymbol{y} \mid \boldsymbol{\lambda} \in \Lambda\}$ constitutes all convex combinations of the objectives in $\Lambda$. The parameters $\boldsymbol{\lambda} \in \Lambda$ can be viewed as weights, or relative preferences, on the objective functions in the scalarized optimization objective $\max_{x \in X} s_\boldsymbol{\lambda}([f_1(x), \dots, f_L(x)])$. Then, the advantage of using scalarization functions is that the solution to maximizing $s_\boldsymbol{\lambda}([f_1(x), \dots, f_L(x)])$, for a fixed value of $\boldsymbol{\lambda}$, is a solution along the PF.

### 2.2 SOFT-HARD UTILITY FUNCTIONS

Many practical applications of multi-objective problems, e.g. engineering design and healthcare, include strict *constraints* on the optimization that define the feasible set of solutions. While these constraints can be directly incorporated into scalarized objectives via penalty methods involving

barrier functions or interior-point methods, the incorporated objectives are often significantly less interpretable for practitioners. Additionally, constraints are often looser, and encode ideal regions for solutions, as opposed to defining feasibility. Again, incorporation into the objective function limits interpretability for a DM and may introduce additional slack variables for optimization. We limit our scope in this capacity to linear constraints of already-defined optimization variables and objective functions of the form $f_\ell(x) \geq \alpha$, for some $\ell \in [L]$. We assert that a myriad of real-world optimization problems have a desirable region for each $f_\ell$, i.e. ideally $f_\ell(x) \geq \alpha_{\ell,S}$ for some $\alpha_{\ell,S}$. Given multiple objectives to optimize, $\alpha_{\ell,S}$ may not be attainable, however, due to external considerations, $f_\ell(x)$ should not drop below some stricter threshold, $\alpha_{\ell,H}$. We rigorously support this assertion through real-world examples and empirical results in Section 5.

We operationalize this notion of constraints via *soft-hard* utility functions, or SHFs, which encode the intuition laid out above. In particular, SHFs transform each $f_\ell$ to a utility space in which (1) $f_\ell$ values under $\alpha_{H,\ell}$ map to $-\infty$, the transformation is (2) concave when input $f_\ell(x) \geq \alpha_{\ell,S}$ (diminishing returns after the soft bound is attained), (3) saturated when input $f_\ell(x) \geq \alpha_{\ell,\tau}$ (to prevent exploding utility values which overwhelm those for the other input dimensions), and (4) the transformation is monotonically increasing in $f_\ell(x)$ to maintain the purpose of maximization.

Here, we present a specific class of SHFs that take the form of a piecewise-linear function. We select this form due to its simplicity, however, functional classes which possess the aforementioned four traits should suffice. Let $\varphi$ denote some objective function with soft-hard bounds $\alpha_S, \alpha_H$, respectively. We define its SHF utility functions as follows:

$$u_\varphi(x) = \begin{cases} 1 + 2\beta \times (\tilde{\alpha}_\tau - \tilde{\alpha}_S) & \varphi(x) \geq \alpha_\tau \\ 1 + 2\beta \times (\tilde{\varphi}(x) - \tilde{\alpha}_S) & \alpha_S < \varphi(x) < \alpha_\tau \\ 1 & \varphi(x) = \alpha_S \\ 2 \times \tilde{\varphi}(x) & \alpha_H < \varphi(x) < \alpha_S \\ 0 & \varphi(x) = \alpha_H \\ -\infty & \varphi(x) < \alpha_H \end{cases} \tag{1}$$

where $\tilde{\varphi}(x)$ and $\tilde{\alpha}$ are the soft-hard bound normalized values [1], $\alpha_\tau$, the saturation point, determines where the utility values begin to saturate (as previously described, to prevent exploding utility values) [2], and $\beta \in [0, 1]$ determines the fraction of the original rate of utility, in $[\alpha_H, \alpha_S]$, obtained within $[\alpha_S, \alpha_\tau]$. An example of $u_\varphi(x)$ for a specific function $\varphi$ is shown in Figure 2.

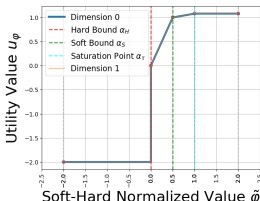

Figure 2: Example of a normalized soft-hard bounded utility function $u$ for the two-dimensional Branin-Currin function. The dashed vertical bars highlight the regions where the normalized values correspond to the hard, soft, and saturation point regions. The utility value associated with points below the hard bound is shown as $-2$ for illustration only. Computationally, we use $-\inf$.

## 2.3 PROBLEM DEFINITION

Given a selected class of scalarization functions $s_\lambda$ parameterized by $\lambda \in \Lambda$ and an SHF utility function as defined above, we wish to elucidate a set of useful points along the PF in the optimization problem

$$\max_{x \in X} s_\lambda(u_f(x))$$

---

[1]Normalization, for value $z$, is performed according to the soft and hard bounds, $\alpha_S$ and $\alpha_H$, respectively, using: $\tilde{z} = ((z - \alpha_H)/(\alpha_S - \alpha_H)) * 0.5$

[2]In our experiments, we determine $\alpha_\tau$ to be $\alpha_H + \zeta(\alpha_S - \alpha_H)$, for $\zeta > 1.0$. We set $\zeta = 2.0$.

where $u_f := [u_{f_1}, \ldots, u_{f_L}]$. Since we only want to select points from the PF, we follow Roijers et al. (2013) and Paria et al. (2019) in assuming that $s_{\boldsymbol{\lambda}}(u_f(x))$ is monotonically increasing in all coordinates $u_{f_\ell}(x)$, which leads to the solution to $\arg\max_{x \in X} s_{\boldsymbol{\lambda}}(u_f(x))$ residing on the PF, for some $\boldsymbol{\lambda}$. We assume the DM has some hidden set of preferences, $\boldsymbol{\lambda}^*$, for which the solution to the optimization problem $\max_{x \in X} s_{\boldsymbol{\lambda}^*}(u_f(x))$ represents the ideal solution on the PF. Therefore, we wish to return a set of Pareto optimal points, $C$, which contains the unknown $c^* = \arg\max_{x \in X} s_{\boldsymbol{\lambda}^*}(u_f(x))$, i.e. the ideal solution that aligns with the DM preferences. Since $\boldsymbol{\lambda}^*$ is unknown to us, we want a set $C$ which is robust against any potential $\boldsymbol{\lambda}^*$, or weight (preference) from the DM. Furthermore, $|C| \leq k$ for some small integer value $k$ to avoid overwhelming the DM with too many choices, which has been shown to decrease choice quality (Diehl, 2005).

As a result, we formulate our general problem of selecting a set of points from the regions defined by the soft and hard bounds as:

$$\max_{C \subseteq X, |C| \leq k} \min_{\boldsymbol{\lambda} \in \Lambda} \left[ \frac{\max_{x \in C} s_{\boldsymbol{\lambda}}(u_f(x))}{\max_{x \in X} s_{\boldsymbol{\lambda}}(u_f(x))} \right] \tag{2}$$

We refer to the right term, $\max_{x \in C} s_{\boldsymbol{\lambda}}(u_f(x)) / \max_{x \in X} s_{\boldsymbol{\lambda}}(u_f(x))$, as the SHF utility ratio. Intuitively, the SHF utility ratio is maximized when the points in $C$ are Pareto optimal and span the high utility regions of the PF, as defined by the SHFs. We use the SHF utility ratio as an evaluation metric for some of our experiments in Section 5.

## 3 STEP 1: DENSE PARETO FRONTIER SAMPLING WITH BAYESIAN OPT.

In this section, we consider the goal of obtaining a dense set of Pareto optimal points. As described earlier, since the DM's preferences, $\boldsymbol{\lambda}^*$, are unknown to us, we wish to obtain a set $D$ which is diverse, high-coverage, and is modeled after the DM-defined SHFs. As is typical in various science and engineering applications, we assume access to some noisy and expensive black-box function – often modeled with a Gaussian process (GP) (Williams & Rasmussen, 1995) as the surrogate function. To achieve that goal, we extend our formulation (2) into a Bayesian setting, assuming a prior $p(\boldsymbol{\lambda})$ with support $\Lambda$ imposed on the set of Pareto optimal values and using the notion of random scalarizations (Paria et al., 2019). In this continuous and stochastic setting, we assume that each of the $\ell \in [L]$ objectives are sampled from known GP priors with a common domain, and produce noisy observations, e.g. $y_\ell = f_\ell(x) + \epsilon_\ell$, where $\epsilon \sim N(0, \sigma_\ell^2)$, $\forall \ell \in [L]$. We optimize over a set of scalarizations weighted by the prior $p(\boldsymbol{\lambda})$.

Our overall aim is still to obtain a set of points $C$ on the PF which are robust against the worst-case potential $\boldsymbol{\lambda}^*$, within the user-defined SHF. For computational feasibility, however, we convert the worst-case into an average-case maximization (Appendix A.1.1). Taking into consideration the aforementioned set of assumptions, we modify the formulation (2) to be the following:

$$\max_{D \subseteq X, |D| \leq k_D} \mathbb{E}_{\boldsymbol{\lambda} \sim p(\boldsymbol{\lambda})} \left[ \frac{\max_{x \in D} s_{\boldsymbol{\lambda}}(u_f(x))}{\max_{x \in X} s_{\boldsymbol{\lambda}}(u_f(x))} \right] \tag{3}$$

**Random Scalarizations.** In this section we describe our sampling-based algorithm to optimize Equation 3. Similar to Paria et al. (2019), we use the notion of random scalarizations to sample a $\boldsymbol{\lambda}$ from $p(\boldsymbol{\lambda})$ at each iteration which is then used to compute a multi-objective acquisition function based on $s_{\boldsymbol{\lambda}}$ and the SHF. The maximizer of the multi-objective acquisition function is then chosen as the next sample input value to be evaluated with the expensive black-box function. The full algorithm, which we refer to as MoSH-Dense, is described in Algorithm 1. We provide formal guarantees proving the lower bound of the SHF utility ratio, which goes to one as $T \to \infty$, and thus providing the dense set $D$ (in Appendix A.3).

## 4 STEP 2: PARETO FRONTIER SPARSIFICATION

We now assume there already exists a dense set of points on the PF, sampled from step 1 with SHFs. As DM validation of such a dense set would be costly, the goal for step 2 is now to sparsify that set

---

**Algorithm 1** MoSH-Dense: Dense Pareto Frontier Sampling Algorithm

---

1: **procedure** MOSH-DENSE
2:     Initialize soft and hard bounds $\{\alpha_{\ell,H}, \alpha_{\ell,S}\} \; \forall \; \ell \in [L]$
3:     Initialize $D^{(0)} = \emptyset$
4:     Initialize $GP_\ell^{(0)} = GP(0, \kappa) \; \forall \; \ell \in [L]$
5:     **for** t = 1 → T **do**
6:         Obtain $\lambda_t \sim p(\lambda)$
7:         $x_t = \arg\max_{x \in X} \text{acq}(u, \lambda_t, x)$       ▷ Details in Appendix A.1
8:         Obtain $y = f(x_t)$
9:         $D^{(t)} = D^{(t-1)} \cup \{(x_t, y)\}$
10:       $GP_\ell^{(t)} = \text{post}(GP_\ell^{t-1}|(x_t, y)) \; \forall \; \ell \in [L]$
11:     **end for**
12:     Return $D^{(T)}$
13: **end procedure**

---

of points to then present to the DM a more navigable set which still maintains as much utility as the dense set. We do so by leveraging the notion of diminishing returns in utility for each additional point the DM validates. This notion is encapsulated by the property of submodularity, which further allows us to design optimization algorithms with strong theoretical guarantees. We use the definition of submodularity first developed in Nemhauser et al. (1978) (see Appendix A.3 for definition).

As a result, we adapt Equation 2 and formulate the sparsification problem as a robust submodular observation selection (RSOS) problem (Krause et al., 2008):

$$\max_{C \subseteq D, |C| \leq k} \min_{\lambda \in \Lambda} \underbrace{\left[ \frac{\max_{x \in C} s_\lambda(u_f(x))}{\max_{x \in D} s_\lambda(u_f(x))} \right]}_{F_\lambda} \tag{4}$$

where $D$ represents the PO dense set obtained from step 1 (Section 3, Algorithm 1) and $C$ is the sparse set of SHF-defined PO points returned to the DM. Equation 4 is not submodular, but can be viewed as a maximin over submodular functions $F_\lambda$ (Lemma 1), studied by Krause et al. (2008).

**Lemma 1.** *For some DM preference value $\lambda$, the set function, which takes as input some SHF-defined set $C$: $\max_{x \in C} s_\lambda(u_f(x)) / \max_{x \in D} s_\lambda(u_f(x))$ is normalized ($F_\lambda(\emptyset) = 0$), monotonic (for all $A \subseteq C \subseteq D, F_\lambda(A) \leq F_\lambda(C)$), and submodular. (See proof in Appendix A.3).*

**Algorithm.** If Equation 4 were submodular, the simple greedy algorithm would provide a near optimal solution (Nemhauser et al., 1978) (theorem in Appendix A.2). Since it is not, the greedy algorithm performs arbitrarily worse than the optimal solution when solving Equation 4, or more generally, problems formulated as RSOS (Krause et al., 2008) - often defined as such:

$$\max_{C \subseteq X, |C| \leq k} \min_i F_i(C) \tag{5}$$

where the goal is to find a set $C$ of observations which is robust against the worst possible objective, $\min_i F_i$, from a set of submodular objectives. As a result, we solve Equation 4 using the Submodular Saturation algorithm, or SATURATE, which provides us with strong theoretical guarantees on the optimality of set $C$ (Krause et al., 2008).

**Theorem 2.** *(Krause et al., 2008) For any integer k, SATURATE finds a solution $C_S$ such that $\min_i F_i(C_S) \geq \max_{|C| \leq k} \min_i F_i(C)$ and $|C_S| \leq \psi k$, for $\psi = 1 + \log(\max_{x \in D} \sum_i F_i(\{x\}))$. The total number of submodular function evaluations is $\mathcal{O}(|D|^2 m \log(m \min_i F_i(D)))$, where $m = |\Lambda|$.*

At a high level, SATURATE first defines a relaxed version of the original RSOS problem, which contains a superset of feasible solutions, and is guaranteed to find solutions to that relaxed version which are at least as informative as the optimal solution, only at a slightly higher cost. Within our context, this enables us to select a subset of points $C$, from the dense set obtained from Step 1, $D$, which achieves the optimal coverage of $\lambda \in \Lambda$ albeit with a slightly greater number of points. The complete algorithm, which we refer to as MoSH-Sparse, is shown in Algorithm 2.

**Algorithm 2** MoSH-Sparse: PF Sparsification

1: **procedure** MoSH-SPARSE($F_1, ..., F_{|\Lambda|}, k, \psi$)
2: $\quad q_{min} = 0; q_{max} = \min_i F_i(D); C_{best} = 0$
3: $\quad$ **while** $(q_{max} - q_{min}) \geq 1 \diagup |\Lambda|$ **do**
4: $\quad\quad q = (q_{min} + q_{max})/2$
5: $\quad\quad$ Define $\bar{F}_q(C) = 1 \diagup |\Lambda| \sum_i \min\{F_i(C), q\}$
6: $\quad\quad \hat{C} = GPC(\bar{F}_q, q)$
7: $\quad\quad$ **if** $|\hat{C}| > \psi k$ **then**
8: $\quad\quad\quad q_{max} = q$
9: $\quad\quad$ **else**
10: $\quad\quad\quad q_{min} = q; C_{best} = \hat{C}$
11:

**Algorithm 3** Greedy Submodular Partial Cover (GPC) Algorithm (Krause et al., 2008)

1: **procedure** GPC($\bar{F}_q, q$)
2: $\quad C = \emptyset$
3: $\quad$ **while** $\bar{F}_q(C) < q$ **do**
4: $\quad\quad$ **foreach** $c \in D \backslash C$ **do** $\delta_c = \bar{F}_q(C \cup \{c\}) - \bar{F}_q(C)$
5: $\quad\quad C = C \cup \{\arg\max_c \delta_c\}$
6:

# 5 EXPERIMENTAL RESULTS

## 5.1 BASELINE METHODOLOGIES

**Step 1: Dense Pareto Frontier Sampling:** We experiment with both synthetic problems and real-world applications and compare our method to other similar Bayesian multi-objective optimization approaches: Expected hypervolume improvement (EHVI) (Emmerich, 2008), ParEGO (Knowles, 2006), Multi-objective Bayesian optimization Using Random Scalarizations (MOBO-RS) (Paria et al., 2019), and random sampling. We compare against MOBO-RS with variations on the scalarization function, Chebyshev and linear, and acquisition function, UCB and Thompson sampling.

**Step 2: Pareto Frontier Sparsification:** We compare our method, MoSH-Sparse, against greedy and random algorithms. The greedy baseline starts with the empty set, and iteratively adds the element $c = \arg\max_{x \in D \backslash C} H(C \cup \{x\})$, where $H = \min_{\boldsymbol{\lambda} \in \Lambda} F_{\boldsymbol{\lambda}}$ for the $F_{\boldsymbol{\lambda}}$ in Equation 4, until some stopping point. For all experiments, additional details and figures are provided in the Appendix.

## 5.2 PERFORMANCE EVALUATION

### 5.2.1 EVALUATION OF STEP 1: DENSE PARETO FRONTIER SAMPLING

As mentioned earlier in Section 3, since the DM's preferences, $\boldsymbol{\lambda}^*$ are unknown to us, we wish to obtain a set D which is (1) diverse, (2) high-coverage, and is (3) modeled after the DM-defined SHFs. We operationalize the three criteria for soft regions ($A_S = [\alpha_{1,S}, +\infty] \times ... \times [\alpha_{L,S}, +\infty]$) and hard regions ($A_H = [\alpha_{1,H}, +\infty] \times ... \times [\alpha_{L,H}, +\infty]$) into four different metrics:

**Soft-Hard Fill Distance**: We seek to measure the diversity of sampled points. Malkomes et al. (2021) measures diversity using the notion of fill distance: FILL($C$, $D$) = $\sup_{x' \in D} \min_{x \in C} \kappa(f(x), f(x'))$ where $C$ is the set of sampled points, $D$ is a full set of precomputed points in the region, and $\kappa(\cdot)$ is the distance metric, typically Euclidean distance. We expand this to include the notion of soft and hard regions: $\upsilon$FILL$_S(C_S, D_S) + (1 - \upsilon)$FILL$_H(C_H, D_H)$, where FILL$_s(C_S, D_S)$ and FILL$_h(C_H, D_H)$ are the fill distances which correspond to the regions defined by the soft bounds and hard bounds, respectively, and $C_S, D_S$ denote the set of points in the soft region, $C \cap A_S$, $D \cap A_S$, (same for hard region). Intuitively, we wish to obtain a diverse sample set which effectively explores both the soft and hard regions, which a higher weighting towards the soft region. We use the $\upsilon$ parameter to control that weighting in our experiments.

**Soft-Hard Positive Samples Ratio**: We seek to measure faithfulness to the implicit constraints set by the SHFs by measuring the ratio of sampled points in the soft and hard regions, defined as: $\upsilon(|C_S| \diagup |C|) + (1 - \upsilon)(|C_H| \diagup |C|)$.

**Soft-Hard Hypervolume**: We seek to measure the coverage of the sampled points by measuring the hypervolume defined by both the soft and hard bounds. We measure the hypervolume of the soft region in the metric space bounded by the PF and the intersection of the soft bounds, $r_S$, $(\alpha_{1,S}, \ldots, \alpha_{L,S})$. The same goes for the hard hypervolume measure.

**Soft Region Distance-Weighted Score**: We seek to explicitly measure faithfulness to the high-utility regions of SHFs, the intersection of the soft bounds, by measuring the density of points. We calculate this using the following: $\sum_{x \in C} 1/\kappa(r_S, f(x))$, where $\kappa$ is a measure of distance, typically Euclidean distance and $r_S$ is defined above.

### 5.2.2 EVALUATION OF STEP 2: PARETO FRONTIER SPARSIFICATION

To evaluate the set of points, $C$, returned to the DM, we simulate a $\boldsymbol{\lambda}^*$ using the heuristic $\boldsymbol{\lambda} = \mathbf{u}/\|\mathbf{u}\|_1$, where $\mathbf{u}_\ell \sim \mathrm{N}(\alpha_{\ell,S}, |\alpha_{\ell,H} - \alpha_{\ell,S}|/3)$. We then compute the SHF utility ratio, right term in Equation 4, for the set of points $C_t$ after each iteration of greedy, random, or MoSH-Dense. The denominator is calculated using $\boldsymbol{\lambda}^*$ with a full set of points, $D$, computed offline.

### 5.3 STEP 1 EXPERIMENTS: SYNTHETIC AND REAL-WORLD APPLICATIONS

### 5.3.1 BRANIN-CURRIN AND ENGINEERING DESIGN PROBLEM: FOUR BAR TRUSS

We leverage the Branin-Currin synthetic two-objective problem provided in the BoTorch framework (Balandat et al., 2020) and performed experiments on multiple variations (Appendix A.4.1). We also evaluate on a MOO engineering design problem, four bar truss, from REPROBLEM (Tanabe & Ishibuchi, 2020), which consists of two objectives and four continuous decision variables, along with a convex PF (CHENG & LI, 1999). The objectives of the problem are to minimize the structural volume and the joint displacement of the four bar truss. The four decision variables determine the length of the four bars, respectively. To demonstrate our method's flexibility, we sample from the following variations: (1) *narrow-mid*, (2) *narrow-bot*, (3) *narrow-top*, (4) *bot-mid*, and (5) *top-mid*. Figure 3 shows the results. Similar to the Branin-Currin experiment, our algorithm matches or surpasses other baselines in all metrics, while sampling at a clearly higher density near the soft region. The other variations' results (in the Appendix) display a similar pattern.

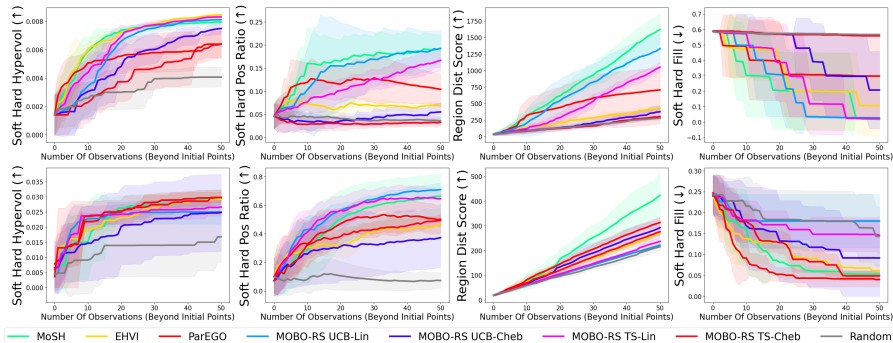

Figure 3: Top row: Four Bar Truss, *narrow-mid*. Bottom row: LLM personalization problem. Plots show the metrics defined in Section 5.2. The mean $\pm$ std. were computed over 6 independent runs.

### 5.3.2 LARGE LANGUAGE MODEL PERSONALIZATION: CONCISE AND INFORMATIVE

We seek to obtain a large language model (LLM) which generates both concise and informative outputs, two directly competing objectives. Rather than fine-tune for both objectives, we leverage proxy tuning, which steers a large pre-trained model, $M$, by using the difference between the predicted logits of an expert model (a smaller, tuned model), $M^+$ and an anti-expert model (the smaller model, un-tuned), $M^-$ (Liu et al., 2024; Mitchell et al., 2023; Shi et al., 2024). We leverage notation from Liu et al. (2024) and obtain the output distribution at time step $t$, conditioned on prompt $x_{<t}$, from the proxy-tuned model, $\tilde{M}$, in a two-objective setting as such: $p_{\tilde{M}}(X_t|x_{<t}) = \mathrm{softmax}[s_M(X_t|x_{<t}) + \sum_{i=1}^2 \theta_i(s_{M_i^+}(X_t|x_{<t}) - s_{M_i^-}(X_t|x_{<t}))]$, where $s_M$, $s_{M_i^+}$, $s_{M_i^-}$ represent the logit scores for each model and $\theta_i$ denotes the input decision variable, the controllable weight applied to the logits difference associated with expert model $i$. For this experiment,

models $M_i^+$, for $i = 0, 1$, are tuned according to the conciseness and informativeness objectives, respectively. By adjusting $\theta_i$ at decoding time, we obtain generated output distributions with varying tradeoffs in conciseness and informativeness. Figure 3 illustrates the results. Across most metrics, our algorithm performs consistently well (moreso than any other baseline) – highlighting the generality of our pipeline. Sample outputs and additional details are provided in Appendix A.4.5.

### 5.3.3 REAL CLINICAL CASE: CERVICAL CANCER BRACHYTHERAPY TREATMENT PLAN

We evaluate our method on treatment planning for a real cervical cancer brachytherapy clinical case. This problem consists of four objectives and three continuous decision variables. The objective are (1) maximize the radiation dosage level to the cancer tumor, and minimize the radiation dosage levels to the (2) bladder, (3) rectum, and (4) bowel. We converted objectives (2)-(4) into maximization objectives. The decision variables are used as inputs to a linear program formulated as an epsilon-constraint method (Deufel et al., 2020). Figure 4 illustrates the plots with the soft-hard performance metrics. We notice that our proposed method surpasses the baselines by a greater amount in this high-dimensional setting, notably the soft-hard hypervolume and soft-hard positive ratio metrics.

### 5.3.4 DEEP LEARNING MODEL SELECTION: FAST AND ACCURATE NEURAL NETWORK

Similar to Hernández-Lobato et al. (2015), we seek to obtain a neural network which minimizes both prediction error and inference time, two competing objectives. We use the MNIST dataset and consider feedforward neural networks with six decision variables (details in Appendix). Figure 4 illustrates the results using the soft-hard metrics. Although our algorithm does not surpass the baselines all four of the metrics, it still performs consistently relatively well in all four (the most consistently well out of the baselines) – in line with the other experimental results as well.

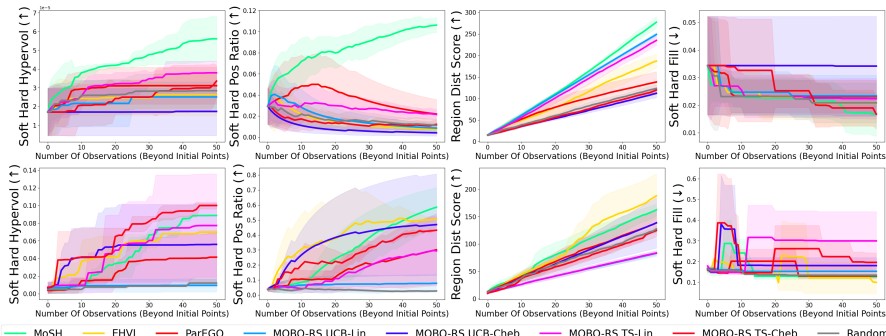

Figure 4: Top row: cervical cancer brachytherapy treatment planning. Bottom row: deep learning model selection, which aims to select a fast and accurate neural network. The plots illustrate the metrics defined in Section 5.2. The mean $\pm$ std. were computed over 6 independent runs.

### 5.4 STEP 2 EXPERIMENTS: SYNTHETIC AND REAL-WORLD APPLICATIONS

We conduct experiments evaluating our baselines on the sparsification of the dense set of points from MoSH-Dense. Figure 5 displays the SHF utility ratio values for each successive point that the DM views, across all four applications. We observe that in all four experimental settings, MoSH-Sparse matches or exceeds the baselines in achieving the overall highest SHF utility ratio. In the brachytherapy setting, MoSH-Sparse achieves the highest SHF utility ratio at the fastest pace, showcasing the effectiveness of our approach in distilling the dense set of points into a useful smaller set.

### 5.5 END-TO-END EXPERIMENTS: SYNTHETIC AND REAL-WORLD APPLICATIONS

We further holistically evaluate our entire two-step process by comparing against all baselines for step (1), using MoSH-Sparse – displayed in Figure 5. We show that our method achieves an over 3% greater SHF utility ratio than the next best one. We further note that our method consistently leads in providing the most utility in all of the experiments (more in Appendix).

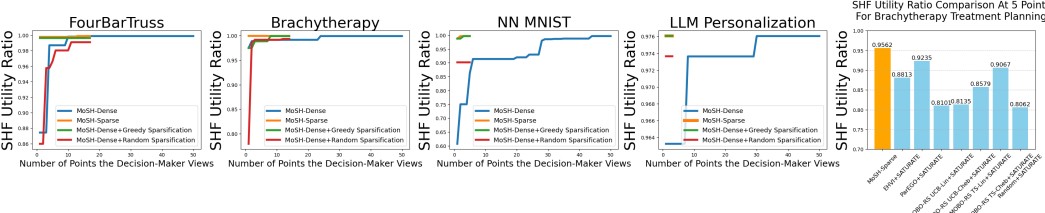

Figure 5: The first four plots evaluate for step 2 of MoSH, illustrating the SHF utility ratio for each successive point that the DM views. The bar plot illustrates the SHF utility ratio obtained by our method compared to the other baselines. Only the dense set, from step 1, changes for each bar.

## 6 RELATED WORKS

**Populating the Pareto Frontier.** The majority of existing MOO works aim to approximate the entire PF, using heuristics such as the maximum hypervolume improvement (Campigotto et al., 2014; Ponweiser et al., 2008; Emmerich, 2008; Picheny, 2015; Hernández-Lobato et al., 2016; Zhang et al., 2009). Others leverage RS of the objective values to attempt to recover the entire PF (Knowles, 2006; Paria et al., 2020; Zhang & Li, 2007; Zhang et al., 2010). Additional works also place a greater emphasis on sparse and diversified PF coverage (Zuluaga et al.; 2016) – more recently, in hard-constrained regions (Malkomes et al., 2021). In contrast, we employ RS in a novel setting which aims to diversely sample a *soft-hard subset* of the PF according to the SHFs.

**MOO Feedback Mechanisms.** Many feedback mechanisms, such as pairwise feedback, have been proposed, albeit not all of which are designed for MOO (Zintgraf et al., 2018; Roijers et al.; Astudillo & Frazier, 2020). Besides pairwise feedback, Hakanen & Knowles (2017) enables the DM to guide the MOO search by allowing them to specify numerical ranges for each of the objectives. Abdolshah et al. (2019) allows for the DM to order objectives by importance, Ozaki et al. (2023) introduced improvement request feedback type for MOO. These methods enable fine-grained MOO control, but our approach uniquely accounts for multiple levels of preferences without needing to specify exact numerical values, which is often psychologically more difficult (Qian et al., 2015).

**Level Set Estimation.** The formulation of objectives as inequality constraints, where the DMs aim to find inputs which satisfy thresholds on the objectives, is related to the topic of level set estimation (LSE) (Gotovos, 2013; Zanette et al., 2018; Iwazaki et al., 2020; Malkomes et al., 2021). In the single-objective setting, Bryan et al. (2005) proposed the straddle heuristic, which was used as part of a GP-based active learning approach for LSE. Although the LSE concept does not easily extend into a MOO setting, Bryan & Schneider (2008) aims to address that by considering the threshold as part of a composite setting, with scalarized objectives. In contrast, our work is native to the MOO setting and additionally incorporates *soft* constraints, which directly leverages the DM's preferences.

## 7 CONCLUSION

In this paper, we introduced a novel setting and formulation for MOO using SHFs, monotonic soft-hard bounded utility functions for each objective, allowing for the DM to impose their preferences via soft and hard bounds. We demonstrated the generality of this setting, and showed how it encompasses the problem of engineering design, treatment planning for cervical cancer brachytherapy, model selection for deep learning, and personalization of LLMs. Within our setting, we then propose a simple two-step process which aims to return a small set of high-utility PO points according to the DM's unknown preferences: (1) dense PF sampling using Bayesian optimization, and (2) sparsification of the PO points from (1) using robust submodular function optimization, which we theoretically show is able to obtain the near-optimal set from (1). Lastly, we propose a set of soft-hard metrics and conduct extensive empirical validations on a variety of applications. We show that, for cervical cancer brachytherapy treatment planning, our approach returns a compact set of treatment plans which offers over 3% greater SHF-defined utility than the next best approach. Among the other diverse experiments, our approach also consistently achieves the leading utility, allowing the DM to reach >99% of their maximum desired utility within validation of 5 points.

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

## A APPENDIX

### A.1 STEP 1: DENSE PARETO FRONTIER SAMPLING WITH BAYESIAN OPT. ADDITIONAL DETAILS

Here we describe the acquisition function used in line 7 of step 1, Algorithm 1. For our experiments, we use the Upper Confidence Bound (UCB) heuristic. We define $\text{acq}(u, \lambda_t, x) = s_{\lambda_t}(u_{\varphi(x)})$ where $\varphi(x) = \mu_t(x) + \sqrt{\beta_t}\sigma_t(x)$ and $\beta_t = \sqrt{0.125 \times \log(2 \times t + 1)}$. For $\beta_t$, we followed the optimal suggestion in Paria et al. (2019).

#### A.1.1 COMPUTATIONAL FEASIBILITY

Equation 3 uses a conversion from the worst-case to an average-case minimization problem. To observe the difficulty of the worst-case minimization problem and how the results may be affected, we performed several experiments which directly solve the following:

$$\max_{D \subseteq X, |D| \leq k_D} \min_{\boldsymbol{\lambda} \in \Lambda} \left[ \frac{\max_{x \in D} s_{\boldsymbol{\lambda}}(u_f(x))}{\max_{x \in X} s_{\boldsymbol{\lambda}}(u_f(x))} \right] \quad (6)$$

We solve Equation 6 using a greedy algorithm over discretized input space $X$ and weight space $\Lambda$ and compare the results.

**Discretization.** To create the finite approximation of the continuous input space, we discretize the domain using a uniform grid. Given input space bounds $[\mathbf{l}, \mathbf{u}] \subset \mathbb{R}^d$ where $\mathbf{l} = [l_1, \ldots, l_d]$ and $\mathbf{u} = [u_1, \ldots, u_d]$ are the lower and upper bounds respectively, we construct a discretized grid as follows:

For each dimension $i \in [d]$, we create an evenly-spaced sequence:

$$\mathcal{X}_i = \{l_i, l_i + \delta, l_i + 2\delta, \ldots, u_i\} \quad (7)$$

where $\delta$ is the step size parameter controlling the granularity of the discretization.

The complete discretized input space $\mathcal{X}_{\text{disc}}$ is then constructed as the Cartesian product:

$$\mathcal{X}_{\text{disc}} = \mathcal{X}_1 \times \mathcal{X}_2 \times \cdots \times \mathcal{X}_d \tag{8}$$

This results in a finite grid of points where each point $\mathbf{x} \in \mathcal{X}_{\text{disc}}$ represents a candidate solution in the original continuous space. The total number of points in the discretized space is $\prod_{i=1}^{d} \left\lceil \frac{u_i - l_i}{\delta} \right\rceil$. The same discretization method is done for $\Lambda$, except, from this grid, we select only those points that lie on the probability simplex, i.e., whose components sum to 1.

**Experiments and Discussions.** We term the baseline algorithms as *discrete-greedy-$\delta$* and primarily experiment with the Branin-Currin objective function (described in Section A.4.1). We show numerical results in Figure 6. In general, computationally, *discrete-greedy-$\delta$* takes longer as $\delta \to 0$. Specifically, *discrete-greedy-0.05* and *discrete-greedy-0.10* take, on average, 20x and 3x longer (in seconds) than MoSH-Dense, respectively, to select the next sample $x_t$ at iteration $t$. This computational disparity is further exacerbated as the size of the input space dimensionality increases. However, as expected, as $\delta \to 0$, the metrics improve and move closer to those of MoSH-Dense. Finally, we note that, as described in Section 4, the greedy algorithm may perform arbitrarily bad when solving Equation 6. We leave for future work exploration of other algorithms which are designed for our setting described in Section 3; notably, access to some noisy and expensive black-box function.

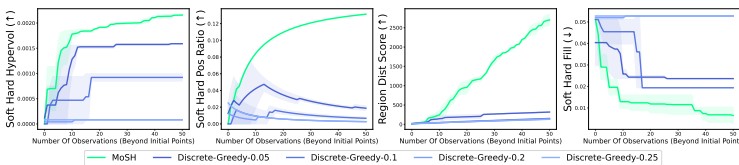

Figure 6: Step (1) results with discrete greedy algorithms which explicitly solve for the max-min problem for MoSH-Dense (Equation 6). The plots compare our proposed method, MoSH-Dense, against several variations of *discrete-greedy-$\delta$* using the metrics defined in Section 5.2.

### A.2 STEP 2: PARETO FRONTIER SPARSIFICATION ADDITIONAL DETAILS

**Theorem 3.** *(Nemhauser et al., 1978) In the case of any normalized, nomotonic submodular function $F$, the set $A_G$ obtained by the greedy algorithm achieves at least a constant fraction $(1 - \frac{1}{e}$ of the objective value obtained by the optimal solution, that is,*

$$F(A_G) \geq (1 - \frac{1}{e}) \max_{|A| \leq k} F(A)$$

### A.3 REQUIRED PROOFS AND DEFINITIONS

**Definition 4** (Pareto dominant). *A solution $x_1 \in X$ is Pareto dominated by another point $x_2 \in X$ if and only if $f_\ell(x_1) \leq f_\ell(x_2) \, \forall \ell \in [L]$ and $\exists \ell \in [L]$ s.t. $f_\ell(x_1) < f_\ell(x_2)$ (Paria et al., 2020).*

**Definition 5** (Submodular). *$F$ is submodular if and only if for all $A \subseteq B \subseteq V$ and $s \in V \backslash B$ it holds that $F(A \cup \{s\}) - F(A) \geq F(B \cup \{s\}) - F(B)$*

**Theorem 6.** *Consider finite sets $\Omega, \Lambda$ and function $f : \Omega \times \Lambda \to \mathbb{R}$. Fix $\lambda \in \Lambda$. Then, given $C \subset \Omega$, the set function*

$$F(C) := \frac{\max_{c \in C} f(c, \lambda)}{\max_{c \in \Omega} f(c, \lambda)}$$

*is submodular.*

*Proof.* Let $X \subseteq Y \subseteq \Omega$ and $x \in \Omega \backslash Y$ (note: it clearly follows that $x \notin X$). Consider $x^* := \max_{c \in \Omega} f(c, \lambda)$. We obtain the following mutually exclusive and exhaustive cases, in which we use the fact that for any sets $A, B$ such that $A \subseteq B$, $\max_{a \in A} f(a) \leq \max_{b \in B} f(b)$ for any function $f$.

- $x^* \in X \subseteq Y$. By definition, $F(X \cup \{x\}) - F(X) = F(Y \cup \{x\}) - F(Y) = 0$.

- $x^* \in Y \backslash X$. By definition, $F(Y \cup \{x\}) - F(Y) = 0$. Clearly $X \subseteq X \cup \{x\}$, thus $F(X \cup \{x\}) - F(X) \geq 0 = F(Y \cup \{x\}) - F(Y)$.

- $x^* \in \Omega \backslash Y$ and $x^* \neq x$.

  - If $F(x, \lambda) \geq \max_{c \in Y} f(c, \lambda)$, then $F(X \cup \{x\}) = F(Y \cup \{x\}) = F(x, \lambda)$.
  - If $F(x, \lambda) \leq \max_{c \in X} f(c, \lambda)$, $F(X \cup \{x\}) = F(X)$ and $F(Y \cup \{x\}) = F(Y)$.
  - Finally, if $\max_{c \in X} f(c, \lambda) \leq F(x, \lambda) \leq \max_{c \in Y} f(c, \lambda)$, then $F(Y \cup \{x\}) = F(Y)$ and $F(X \cup \{x\}) \geq F(X)$.

  Combining with the fact since $X \subseteq Y$, $F(X) \leq F(Y)$, in all of the above sub-cases, we indeed have $F(X \cup \{x\}) - F(X) \geq F(Y \cup \{x\}) - F(Y)$.

- $x^* = x$. As $F(X \cup \{x\}) = F(Y \cup \{x\})$, and $F(X) \leq F(Y)$, we automatically get $F(X \cup \{x\}) - F(X) \geq F(Y \cup \{x\}) - F(Y)$.

Therefore, we satisfy the definition of submodularity. $\qquad\square$

**Definition 7** (Instantaneous SHF Regret)**.** $r(x_t, \lambda_t) = 1 - \dfrac{s_{\lambda_t}(u_f(x_t))}{\max_{x \in X} s_{\lambda_t}(u_f(x_t))}$

**Definition 8** (Cumulative SHF Regret)**.** $R_C(T) = \sum_{t=1}^{T} r(x_t, \lambda_t)$

**Definition 9** (Bayes SHF Regret and Utility Ratio)**.** $R_B(T) = \mathbb{E}_{\lambda \sim p(\lambda)}[1 - \dfrac{\max_{x \in D_T} s_\lambda(u_f(x))}{\max_{x \in X} s_\lambda(u_f(x))}]$

where $D_T = \{x_t\}_{t=1}^{T}$. Additionally, $U_B(T) = \mathbb{E}_{\lambda \sim p(\lambda)}[\dfrac{\max_{x \in D^T} s_\lambda(u_f(x))}{\max_{x \in X} s_\lambda(u_f(x))}]$, where $U_B(T)$ is the Bayes SHF Utility Ratio after $T$ iterations.

**Definition 10** (Expected Bayes SHF Regret and Expected Cumulative SHF Regret)**.** *Similar to (Paria et al., 2020), $\mathbb{E}R_B(T)$ is the expected Bayes SHF Regret, with the expectation being taken over $f$, noise $\epsilon$, and other sources of randomness. Likewise, $\mathbb{E}R_C(T)$ is the expected Cumulative SHF Regret, with the expectation being taken over $f$, noise $\epsilon$, and $\lambda_t$.*

**Definition 11** (Maximum Information Gain)**.** *We leverage this definition from (Paria et al., 2020). The maximum information gain after $T$ observations measures the notion of information gained about random process $f$ after observing some set of points $A$, and is defined as:*

$$\gamma_T = \max_{A \subset X : |A| = T} I(y_A; f) \qquad (9)$$

**Definition 12** (Lipschitz Conditions)**.** *We assume the following is $M_\lambda$-Lipschitz in the $\ell_1$-norm for all $\lambda \in \Lambda$,*

$$\left| \frac{s_\lambda(y_1)}{\max_{y \in Im(u_f)} s_\lambda(y)} - \frac{s_\lambda(y_2)}{\max_{y \in Im(u_f)} s_\lambda(y)} \right| \leq M_\lambda \|y_1 - y_2\|_1 \qquad (10)$$

*where $y \in \mathbb{R}^L$ corresponds to $u_f(x)$.*

**Definition 13** (Lipschitz Condition)**.** *We assume the following is $J$-Lipschitz in $\lambda$ for all $y \in \mathbb{R}^L$.*

$$\left| \frac{s_{\lambda_1}(y)}{\max_{y' \in Im(u_f)} s_{\lambda_1}(y')} - \frac{s_{\lambda_2}(y)}{\max_{y' \in Im(u_f)} s_{\lambda_2}(y')} \right| \leq J \|\lambda_1 - \lambda_2\|_1 \qquad (11)$$

**Definition 14** (Regret Bounds)**.** *We follow similar notation as in (Paria et al., 2020). Assume that $\forall \ell \in [L], t \in [T], x \in \mathcal{X}$, each objective $f_\ell(x)$ follows a Gaussian distribution with marginal variances upper bounded by 1, and the observation noise $\epsilon_{t\ell} \sim \mathcal{N}(0, \sigma_\ell^2)$ is drawn independently of everything else. We assume upper bounds $M_\lambda \leq M$, $\sigma_\ell^2 \leq \sigma^2$, $\gamma_{T\ell} \leq \gamma_T$, where $\gamma_{T\ell}$ is the maximum information gain for the $\ell$th objective. We assume $\mathcal{X} \subseteq [0, 1]^d$. Furthermore, let $x_t^* = \arg\max_{x \in X} s_{\lambda_t}(u_f(x))$. We denote by $U_t(\lambda, x) = s_\lambda(u_{\varphi(x)})$ where $\varphi(x) = \mu_t(x) + \sqrt{\beta_t} \sigma_t(x)$. Finally, the history until T-1 is denoted as $\mathcal{H}_t$, i.e. $\{(x_t, y_t, \lambda_t)\}_{t=1}^{T-1}$.*

**Theorem 15.** *The expected cumulative SHF regret for MoSH-Dense after T observations can be upper bounded for both Upper Confidence Bound and Thompson Sampling as,*

$$\mathbb{E}R_C(T) = O(M[\frac{L^2 T d \gamma_T \ln T}{\ln(1 + \sigma^{-2})}]^{1/2})$$

*Proof.*

$$\mathbb{E}R_C(T) = \mathbb{E}\left[\sum_{t=1}^{T}\left(1 - \frac{s_{\lambda_t}(u_f(x_t))}{\max_{x \in X} s_{\lambda_t}(u_f(x))}\right)\right]$$

$$= \mathbb{E}\left[\sum_{t=1}^{T}\left(\frac{\max_{x \in X} s_{\lambda_t}(u_f(x))}{\max_{x \in X} s_{\lambda_t}(u_f(x))} - \frac{s_{\lambda_t}(u_f(x_t))}{\max_{x \in X} s_{\lambda_t}(u_f(x))}\right)\right]$$

Using Lemma 5 from Paria et al. (2020), we have the following decomposition for UCB:

$$\mathbb{E}R_C(T) \leq \mathbb{E}\left[\underbrace{\sum_{t=1}^{T} \frac{U_t(\lambda_t, x_t)}{\max_{x \in X} s_{\lambda_t}(u_f(x))} - \frac{s_{\lambda_t}(u_f(x_t))}{\max_{x \in X} s_{\lambda_t}(u_f(x))}}_{B1}\right] +$$

$$\mathbb{E}\left[\underbrace{\sum_{t=1}^{T} \frac{s_{\lambda_t}(u_f([x_t^*]_t))}{\max_{x \in X} s_{\lambda_t}(u_f(x))} - \frac{U_t(\lambda_t, [x_t^*]_t)}{\max_{x \in X} s_{\lambda_t}(u_f(x))}}_{B2}\right] +$$

$$\mathbb{E}\left[\underbrace{\sum_{t=1}^{T} \frac{s_{\lambda_t}(u_f(x_t^*))}{\max_{x \in X} s_{\lambda_t}(u_f(x))} - \frac{s_{\lambda_t}(u_f([x_t^*]_t))}{\max_{x \in X} s_{\lambda_t}(u_f(x))}}_{B3}\right]$$

By using Definition 12, we extend Lemma 3 from (Paria et al., 2020) and obtain the following:

**Lemma 16.** $\mathbb{E}\left[\sum_{t=1}^{T} \frac{U_t(\lambda_t, x_t)}{\max_{x \in X} s_{\lambda_t}(u_f(x))} - \frac{s_{\lambda_t}(u_f(x_t))}{\max_{x \in X} s_{\lambda_t}(u_f(x))}\right] \leq$
$\mathbb{E}\left[(L\beta_T \sum_{t=1}^{T} M_{\boldsymbol{\lambda}_t}^2)^{1/2}(\sum_{\ell=1}^{L} \frac{\gamma_{T\ell}}{\ln(1 + \sigma_\ell^{-2})})^{1/2}\right] + \frac{\pi^2}{6} \frac{L\mathbb{E}[M_{\boldsymbol{\lambda}}]}{|X|}$

By using Definition 12, we extend Lemma 2 from (Paria et al., 2020) and obtain the following:

**Lemma 17.** $\mathbb{E}\left[\sum_{t=1}^{T} \frac{s_{\lambda_t}(u_f(x_t^*))}{\max_{x \in X} s_{\lambda_t}(u_f(x))} - \frac{U_t(\lambda_t, x_t^*)}{\max_{x \in X} s_{\lambda_t}(u_f(x))} - \right] \leq \frac{\pi^2}{6}\mathbb{E}[M_{\boldsymbol{\lambda}}]L$

By using Definition 12, we extend Equation (17) from (Paria et al., 2020) and obtain the following:

**Lemma 18.** $\mathbb{E}\left[|\frac{s_{\lambda}(u_f(x))}{\max_{x \in X} s_{\lambda_t}(u_f(x))} - \frac{s_{\lambda}(u_f([x]_t))}{\max_{x \in X} s_{\lambda_t}(u_f(x))}|\right] \leq L\mathbb{E}[M_{\boldsymbol{\lambda}}]\frac{1}{t^2}$

Finally, we use Lemma 16 to bound B1, Lemma 17 to bound B2, Lemma 18 to bound B3, and obtain:

$$\mathbb{E}R_C(T) \leq C_1 L\mathbb{E}[M_{\boldsymbol{\lambda}}] + C_2\bar{M}_{\boldsymbol{\lambda}}\left(LT(d\ln T + d\ln d)\sum_{\ell=1}^{L} \frac{\gamma_{T\ell}}{\ln(1 + \sigma_\ell^{-2})}\right)^{1/2} \tag{12}$$

which converges to 0 as $T \to \infty$. $\square$

**Theorem 19.** *The expected Bayes SHF regret can be upper bounded as:*

$$\mathbb{E}R_B(T) \leq \frac{1}{T}\mathbb{E}R_C(T) + o(1)$$

*As a result, showing that the SHF Utility Ratio converges to 1 as $T \to \infty$ in MoSH-Dense.*

*Proof.* We assume that $\Lambda$ is a bounded subset of a normed linear space. Following the approach in Paria et al. (2020), we begin by relating the sampling distribution to the empirical distribution. Let $\hat{p}$ denote the empirical distribution corresponding to the samples $\{\lambda_t\}_{t=1}^T$. Consider the Wasserstein (Earth Mover's) distance between the sampling distribution $p(\lambda)$ and $\hat{p}$:

$$W_1(p, \hat{p}) = \inf_q \{\mathbb{E}_q \|X - Y\|_1 : q(X) = p, q(Y) = \hat{p}\} \tag{13}$$

where $q$ is a joint distribution on random variables $X, Y$ with marginals $p$ and $\hat{p}$ respectively.

We can then use Definition 13 to bound the following in the SHF setting:

$$\frac{1}{T} \sum_{t=1}^T (1 - \frac{s_{\lambda_t}(u_f(x_t))}{\max_{x \in X} s_{\lambda_t}(u_f(x))}) - \mathbb{E}\left[1 - \frac{\max_{x \in D} s_{\lambda_t}(u_f(x_t))}{\max_{x \in X} s_{\lambda_t}(u_f(x))}\right]$$

$$\geq \frac{1}{T} \sum_{t=1}^T (1 - \frac{\max_{x \in D} s_{\lambda_t}(u_f(x))}{\max_{x \in X} s_{\lambda_t}(u_f(x))}) - \mathbb{E}\left[1 - \frac{\max_{x \in D} s_{\lambda_t}(u_f(x))}{\max_{x \in X} s_{\lambda_t}(u_f(x))}\right]$$

$$\geq \mathbb{E}_{q(Z,Y)}\left[(1 - \frac{\max_{x \in D} s_Y(u_f(x))}{\max_{x \in X} s_Y(u_f(x))}) - (1 - \frac{\max_{x \in D} s_Z(u_f(x))}{\max_{x \in X} s_Z(u_f(x))})\right]$$

$$\geq -\mathbb{E}_{q(Z,Y)}\{J\|Z - Y\|_1\}$$

Taking the infimum with respect to $q$ and expectation with respect to the history $\mathcal{H}_t$:

$$\mathbb{E}\left[\frac{1}{T} \sum_{t=1}^T (1 - \frac{s_{\lambda_t}(u_f(x_t))}{\max_{x \in X} s_{\lambda_t}(u_f(x))})\right] - \mathbb{E}\left[1 - \frac{\max_{x \in D} s_{\lambda_t}(u_f(x_t))}{\max_{x \in X} s_{\lambda_t}(u_f(x))}\right] \geq -J\mathbb{E}W_1(p, \hat{p}) \tag{14}$$

Using the fact that $\mathbb{E}[\max_{x \in \mathcal{X}} s_\lambda(u_f(x))] = \mathbb{E}[\max_{x \in \mathcal{X}} s_{\lambda_t}(u_f(x))]$, we obtain:

$$\mathbb{E}R_B(T) \leq \frac{1}{T}\mathbb{E}R_C(T) + J\mathbb{E}W_1(p, \hat{p}) \tag{15}$$

By Theorem 15 and results from Paria et al. (2020), the first term converges to zero at rate $O^*(T^{-1/2})^4$ and the second term converges to zero at rate $O^*(T^{-1/D})$ for $D \geq 2$ (Canas & Rosasco, 2012), where $D$ is the dimension of $\Lambda$. As a result, $\mathbb{E}R_B(T) \to 0$ as $T \to \infty$. Since $R_B(T)$ is the inverse of $U_B(T)$, from Definition 9, $\mathbb{E}U_B(T) \to 1$ as $T \to \infty$. $\square$

**Lemma 20.** *For a fixed $\boldsymbol{\lambda} \in \Lambda$, the augmented Chebyshev scalarization function $s_{\boldsymbol{\lambda}}(y) = -\max_{\ell \in [L]}\{\boldsymbol{\lambda}_\ell |y_\ell - z_\ell^*|\} - \gamma \sum_{\ell=1}^L |y_\ell - z_\ell^*|$, as described in Section A.5.1, satisfies the assumption in Definition 12.*

*Proof.* Let $\boldsymbol{\lambda} \in \Lambda$. Recall $\Lambda := \Delta^L$, thus is bounded. Furthermore, $\text{Im}(u_f)$ is bounded when $f(x) \geq \alpha_H$. First, we demonstrate that $s_{\boldsymbol{\lambda}}(y)$ is Lipschitz w.r.t. $y$. For $\ell \in [L]$,

$$\frac{\partial s_{\boldsymbol{\lambda}}(y)}{\partial y_\ell} = \begin{cases} -\lambda_{\ell^*}, & y_{\ell^*} > z_{\ell^*}^*, \ell = \ell^* \\ \lambda_{\ell^*}, & y_{\ell^*} < z_{\ell^*}^*, \ell = \ell^* \\ 0, & \ell \neq \ell^* \end{cases}\Bigg|_{\ell^* := \arg\max_\ell\{\boldsymbol{\lambda}_\ell |y_\ell - z_\ell^*|\}} - \gamma \begin{cases} 1 & y_\ell > z_\ell^* \\ -1 & y_\ell < z_\ell^* \end{cases}$$

$$\implies \left\|\frac{\partial s_{\boldsymbol{\lambda}}(y)}{\partial y}\right\| \leq \lambda_{\ell^*}\Bigg|_{\ell^* := \arg\max_\ell\{\boldsymbol{\lambda}_\ell |y_\ell - z_\ell^*|\}} + L$$

As the partial derivative is bounded w.r.t. $y$, by Mean Value Theorem (MVT) the scalarization function is Lipschitz w.r.t. $y$. Thus, there exists some constant $C_{\boldsymbol{\lambda}}$, such that for all $y_1, y_2 \in \text{Im}(u_f)$,

$$|s_{\boldsymbol{\lambda}}(y_1) - s_{\boldsymbol{\lambda}}(y_2)| \leq C_{\boldsymbol{\lambda}}\|y_1 - y_2\|$$

Note that for fixed $\boldsymbol{\lambda}$, $\max_{y \in \text{Im}(u_f)} s_{\boldsymbol{\lambda}}(y)$ is a constant. Using that fact, and the equation above, Definition 12 follows. $\square$

**Lemma 21.** *The augmented Chebyshev scalarization function $s_{\boldsymbol{\lambda}}(y) = -\max_{\ell \in [L]}\{\boldsymbol{\lambda}_\ell |y_\ell - z_\ell^*|\} - \gamma \sum_{\ell=1}^{L} |y_\ell - z_\ell^*|$, as described in Section A.5.1, satisfies the assumption in Definition 13 when $y \in S$ such that $S$ bounded and $S \subseteq Im(u_f)$.*

*Proof.* We assume the following: $y \in S \backslash \{-\infty\}$ and $\exists \ell \in [L]$ s.t. $y_\ell = M$, where $M$ is the upper bound of the SHF. Intuitively, this assumes the soft bounds are placed such that their intersection, i.e. $(\alpha_\ell, ..., \alpha_L)$, is within the Pareto frontier. First, we apply Danskin's Theorem (Danskin, 1966) to $\max_{y \in Im(u_f)} s_{\boldsymbol{\lambda}}(y)$. If $s_{\boldsymbol{\lambda}}(y)$ is convex in $\boldsymbol{\lambda}$ for all $y \in Im(u_f)$, and at a given $\lambda_o \in \Lambda$, $\exists$ a unique maximizer $y_o \in Im(u_f)$ then $s_{\boldsymbol{\lambda}}(y)$ is differentiable w.r.t. $\boldsymbol{\lambda}$ at $\boldsymbol{\lambda}_o$ with derivative $\dfrac{\partial s_{\boldsymbol{\lambda}}(y_o)}{\partial \boldsymbol{\lambda}}\bigg|_{\boldsymbol{\lambda}=\boldsymbol{\lambda}_o}$.

As a result, if $y_o$ is the unique maximizer at $\boldsymbol{\lambda}_o$, then

$$\left.\frac{\partial s_{\boldsymbol{\lambda}}(y_o)}{\partial \boldsymbol{\lambda}}\right|_{\boldsymbol{\lambda}=\boldsymbol{\lambda}_o} = \left.\left(|y_{o,\ell^*} - z_{\ell^*}^*|\right)\right|_{\ell^*=\arg\max_\ell [\lambda_\ell |y'_\ell - z_\ell^*|]}$$

Then, we demonstrate that $s_{\boldsymbol{\lambda}}(y')/\max_{y \in Im(u_f)} s_{\boldsymbol{\lambda}}(y)$ is Lipschitz w.r.t. $\boldsymbol{\lambda}$. To conserve space, we denote $\max_{y \in Im(u_f)} s_{\boldsymbol{\lambda}}(y)$ with $A$.

$$\left.\frac{\partial}{\partial \boldsymbol{\lambda}}\left[\frac{s_{\boldsymbol{\lambda}}(y')}{A}\right]\right|_{\boldsymbol{\lambda}=\boldsymbol{\lambda}_o} = \frac{\left[(A)\left(|y'_{\ell^*} - z_{\ell^*}^*|\right)\Big|_{\ell^*=\arg\max_\ell [\lambda_\ell |y'_\ell - z_\ell^*|]}\right]}{[A]^2} -$$

$$\frac{\left[(s_{\boldsymbol{\lambda}}(y'))\left(|y_{o,\ell^*} - z_{\ell^*}^*|\right)\Big|_{\ell^*=\arg\max_\ell [\lambda_\ell |y'_\ell - z_\ell^*|]}\right]}{[A]^2}$$

$$= \underbrace{\frac{\left[\left(|y'_{\ell^*} - z_{\ell^*}^*|\right)\Big|_{\ell^*=\arg\max_\ell [\lambda_\ell |y'_\ell - z_\ell^*|]}\right]}{[A]}}_{C1} -$$

$$\underbrace{\frac{\left[(s_{\boldsymbol{\lambda}}(y'))\left(|y_{o,\ell^*} - z_{\ell^*}^*|\right)\Big|_{\ell^*=\arg\max_\ell [\lambda_\ell |y'_\ell - z_\ell^*|]}\right]}{[A]^2}}_{C2}$$

The numerator for term C2 is bounded since $s_{\boldsymbol{\lambda}}$ is in a bounded space. The numerator for C1 is upper bounded since $y$, our SHF, is an upper-bounded function.

The denominator values for C1 and C2 are both lower-bounded: since $\boldsymbol{\lambda}$ lies on the probability simplex, there must be a $\lambda_{\ell^\dagger} \geq 1/L$ for some $\ell^\dagger \in L$, by the pigeonhole principle. Since we are maximizing over $y \in Im(u_f)$, there must be a $y^\dagger$ at the upper bound of $Im(u_f)$, which we denote as $M$. As a result, As a result, $\max_{y \in Im(u_f)} s_{\boldsymbol{\lambda}}(y) \geq M/L$. Since the denominators of both C1 and C2 are lower-bounded, the overall partial derivative is bounded w.r.t. $\boldsymbol{\lambda}$.

Since the overall partial derivative is bounded w.r.t. $\boldsymbol{\lambda}$, by MVT $s_{\boldsymbol{\lambda}}(y')/\max_{y \in Im(u_f)} s_{\boldsymbol{\lambda}}(y)$ is Lipschitz w.r.t. $\boldsymbol{\lambda}$. Thus, there exists some constant $J$ such that for all $\boldsymbol{\lambda}_1, \boldsymbol{\lambda}_2 \in \Lambda$

$$\left|\frac{s_{\boldsymbol{\lambda}_1}(y)}{\max_{y' \in Im(u_f)} s_{\boldsymbol{\lambda}_1}(y')} - \frac{s_{\boldsymbol{\lambda}_2}(y)}{\max_{y' \in Im(u_f)} s_{\boldsymbol{\lambda}_2}(y')}\right| \leq J\|\boldsymbol{\lambda}_1 - \boldsymbol{\lambda}_2\|_1$$

$\square$

### A.4 ADDITIONAL EXPERIMENTAL RESULTS

#### A.4.1 SYNTHETIC TWO-OBJECTIVE FUNCTION: BRANIN-CURRIN

We leverage the Branin-Currin synthetic two-objective optimization problem provided in the BoTorch framework (Balandat et al., 2020), which has a mapping of $[0,1]^2 \to \mathbb{R}^2$. To demonstrate our method's flexibility in accommodating various configurations, we sample from the following variations : (1) *complete-mid*, where the hard bounds cover the *complete* Pareto frontier and the soft bounds are in the *middle* of the hard region (2) *complete-top*, (3) *complete-bot*, (4) *top-mid*, and (5) *bot-mid*. Figures 7, 8, 9, 10, 11 illustrate the plots with the performance metrics defined in Section 5.2. We note that EHVI, although superior in some metrics, is extremely computationally demanding for higher dimensions. Overall, we observe that our algorithm generally matches or surpasses other baselines in all four metrics, with sampling a much higher density near the soft region. The figures for the other configurations display a similar pattern.

**Configurations (normalized to [0,1])** : $\{\alpha_{0,S}, \alpha_{0,H}, \alpha_{1,S}, \alpha_{1,H}\}$

1. *Complete-Mid*: $\{0.988, 0.943, 0.856, 0.618\}$
2. *Complete-Top*: $\{0.969, 0.943, 0.935, 0.618\}$
3. *Complete-Bot*: $\{0.998, 0.943, 0.697, 0.618\}$
4. *Top-Mid*: $\{0.969, 0.940, 0.915, 0.856\}$
5. *Bot-Mid*: $\{0.996, 0.975, 0.737, 0.658\}$

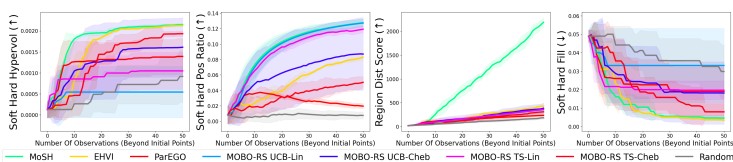

Figure 7: *Complete-Mid* configuration for the Branin-Currin synthetic two-objective function. Results are plotted using the metrics defined in Section 5.2.

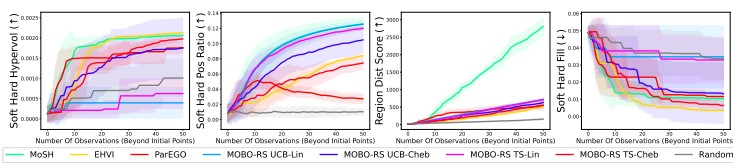

Figure 8: *Complete-Top* configuration for the Branin-Currin synthetic two-objective function. Results are plotted using the metrics defined in Section 5.2.

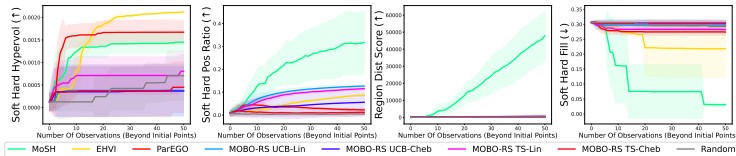

Figure 9: *Complete-Bot* configuration for the Branin-Currin synthetic two-objective function. Results are plotted using the metrics defined in Section 5.2.

#### A.4.2 ENGINEERING DESIGN PROBLEM: FOUR BAR TRUSS

**Configurations (normalized to [0,1])** : $\{\alpha_{0,S}, \alpha_{0,H}, \alpha_{1,S}, \alpha_{1,H}\}$

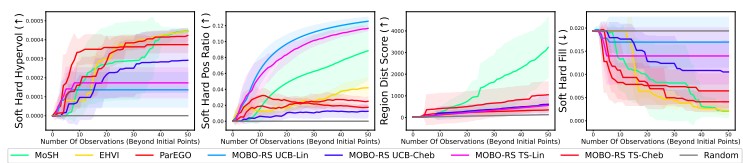

Figure 10: *Top-Mid* configuration for the Branin-Currin synthetic two-objective function. Results are plotted using the metrics defined in Section 5.2.

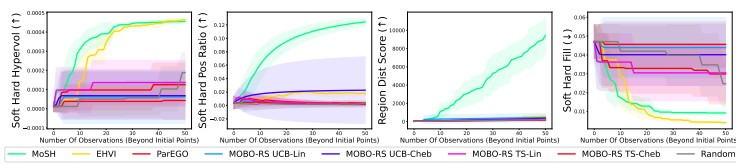

Figure 11: *Bot-Mid* configuration for the Branin-Currin synthetic two-objective function. Results are plotted using the metrics defined in Section 5.2.

1. *Narrow-Mid*: {0.62, 0.45, 0.72, 0.55}
2. *Narrow-Bot*: {0.70, 0.45, 0.65, 0.55}
3. *Narrow-Top*: {0.55, 0.45, 0.78, 0.55}
4. *Bot-Mid*: {0.86, 0.70, 0.48, 0.25}
5. *Top-Mid*: {0.43, 0.20, 0.85, 0.70}

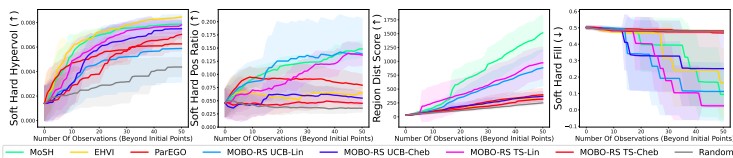

Figure 12: *Narrow-Bot* configuration for the Four Bar Truss engineering design two-objective function. Results are plotted using the metrics defined in Section 5.2.

### A.4.3 REAL CLINICAL CASE: CERVICAL CANCER BRACHYTHERAPY TREATMENT PLANNING

The configuration used was: $\{\alpha_{0,S} = 0.95, \alpha_{0,H} = 0.90, \alpha_{1,S} = 513, \alpha_{1,H} = 601, \alpha_{2,S} = 352, \alpha_{2,H} = 464, \alpha_{3,S} = 411, \alpha_{3,S} = 464\}$, where the objectives correspond to $PTV_{V700}$, $Bladder_{D2cc}$, $Rectum_{D2cc}$, and $Bowel_{D2cc}$, ordered. Before doing the experiment, all of the values were normalized to [0,1] and converted to maximization.

### A.4.4 DEEP LEARNING MODEL SELECTION: FAST AND ACCURATE NEURAL NETWORK

We used the following decision space: number of hidden units per layer ([50, 300]), number of layers ([1, 3]), learning rate ([0, 0.1]), dropout amount ([0.4, 0.6]), l1 regularization ([0, 0.05]), and l2 regularization ([0, 0.05]). We trained each of the neural networks for 100 epochs and converted both of the objectives such that its a maximization problem.

### A.4.5 LARGE LANGUAGE MODEL PERSONALIZATION: CONCISE AND INFORMATIVE

We leveraged models from the TÜLU-2 suite of families for our experiments: TÜLU-2-13B for the large pre-trained model, $M$, and TÜLU-2-DPO-7B for the expert and anti-expert models, $M^+$ and $M^-$(Ivison et al., 2023). $M_1^+$ and $M_2^+$ were tuned using direct preference optimization (DPO) with

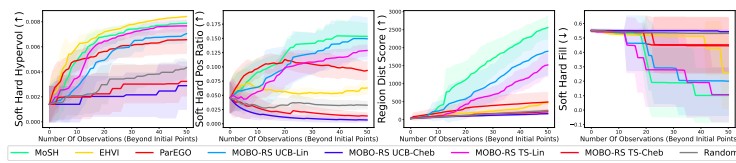

Figure 13: *Narrow-Top* configuration for the Four Bar Truss engineering design two-objective function. Results are plotted using the metrics defined in Section 5.2.

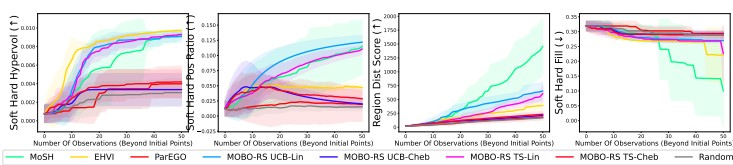

Figure 14: *Bot-Mid* configuration for the Four Bar Truss engineering design two-objective function. Results are plotted using the metrics defined in Section 5.2.

the preference datasets corresponding to the conciseness and informativeness dimensions, respectively, from Jang et al. (2023); Rafailov et al. (2024). We used the following decision space: $\theta_1$ ([0.0, 1.5]) and $\theta_2$ ([0.0, 1.5]). To measure the conciseness dimension, we calculated the number of characters in the output response. To measure the informativeness dimension, we adapted the prompt from Mitchell et al. (2023) and used GPT-4 to provide a measure between 0 and 100:

---

**GPT-4 Informativeness Evaluation Prompt**

"Score the following chatbot response to the given query on a continual scale from 0 (worst) to 100 (best), where a score of 0 means the response lacks information and provides very little information, and a score of 100 means the response is very informative and provides additional background which will be useful for the user.
QUERY: {{the query}}
RESPONSE: {{the response to evaluate}}
USE THE ANSWER FORMAT:
Reason: {analysis of query and response}
Informativeness score: {a single number 0-100, nothing else}"

---

Finally, each of the objectives were normalized and convered to maximization and we used the following five prompts for the experiment and averaged the results:

1. "Give three tips for staying healthy."

2. "What are the three primary colors?"

3. "Describe the structure of an atom."

4. "Rewrite the following instructions so that they are simpler and easier to understand. Login to the website, input your username and password, then click the "Continue" button."

5. "How can we reduce air pollution?"

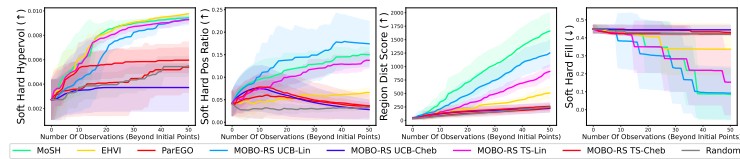

Figure 15: *Top-Mid* configuration for the Four Bar Truss engineering design two-objective function. Results are plotted using the metrics defined in Section 5.2.

---

**Sample LLM Prompt and Responses**

Prompt: Give three tips for staying healthy.

**Response 1**

▷ $\theta_1, \theta_2$: (1.07, 0.00) which corresponds to high conciseness and low informativeness, respectively
Response:

1. Maintain a healthy diet with plenty of fruits and vegetables.

2. Get regular exercise, at least 30 minutes a day.

3. Practice good hygiene and get enough sleep.

▷ GPT-4 Evaluated Informativeness: 50

**Response 2**

▷ $\theta_1, \theta_2$: (1.07, 0.58) which corresponds to high conciseness and medium informativeness, respectively
Response:

1. Maintaining a healthy diet with a balanced intake of nutrients

2. Regular physical activity, at least 30 minutes a day.

3. Getting enough sleep, aim for 7-8 hours per night. (4 in case of power nap). bonus: limit stress. (5th tip: stay hydrated) (6th tip: practice good hygiene) (7th tip: limit consumption of processed foods)

▷ GPT-4 Evaluated Informativeness: 70

---

### A.4.6    STEP 2 EXPERIMENTS: SYNTHETIC AND REAL-WORLD APPLICATIONS

For Figure 5, each successive point that the DM views is determined by the sampling order of the points. Figure 16 shows the evaluations for step (2) on the Branin-Currin synthetic function.

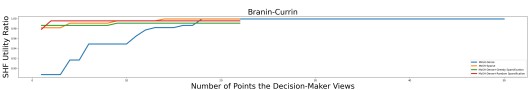

Figure 16: Step (2) evaluation on the Branin-Currin synthetic function, illustrating the SHF utility ratio for each successive point that the DM views.

### A.4.7    END-TO-END EXPERIMENTS: SYNTHETIC AND REAL-WORLD APPLICATIONS

We conducted and end-to-end evaluation of our method, MoSH, by varying the dense set of points, from step 1, provided to SATURATE. Figures 17, 18, 19, and 20 display bar plots representing the maximum SHF utility ratio obtained after the DM views 5 points, for each experiment. For algorithms which select less than 5 points, we use the SHF utility ratio value of the last point.

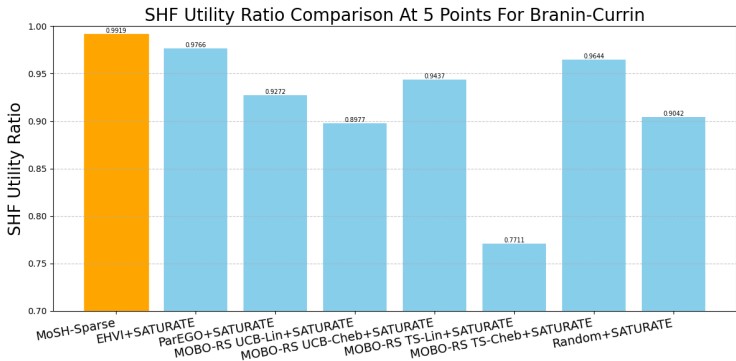

Figure 17: Bar plot illustrating the SHF utility ratio obtained by our method, MoSH-Sparse, compared to the other baselines on the Branin-Currin synthetic function. Only the dense set, from step 1, changes for each bar.

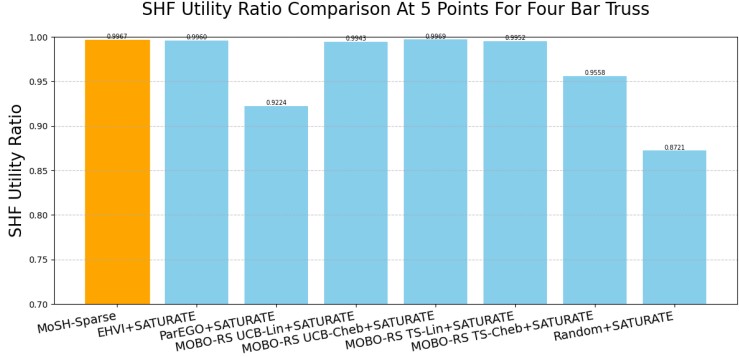

Figure 18: Bar plot illustrating the SHF utility ratio obtained by our method, MoSH-Sparse, compared to the other baselines on the Four Bar Truss application. Only the dense set, from step 1, changes for each bar.

## A.5 EXPERIMENTAL SETUP DETAILS

### A.5.1 BASELINE METHODOLOGIES

**Step 1: Dense Pareto Frontier Sampling.** Although none of the baseline algorithms are inherently designed for our setting with SHFs, we aimed to make the comparisons as fair as possible by using a heuristic, similar to what was described in (Paria et al., 2019), to determine the weight distribution $p(\lambda)$. In short, we used the heuristic $\lambda = \mathbf{u} / \|\mathbf{u}\|_1$, where $\mathbf{u}_l \sim \mathrm{N}(\alpha_{\ell,S}, |\alpha_{\ell,H} - \alpha_{\ell,S}|/3)$, in order to roughly mimic the gradually decreasing weight between the soft bounds $\alpha_{\ell,S}$ and the hard bounds $\alpha_{\ell,H}, \forall \ell \in [L]$.

For our proposed method (MoSH), we use the augmented Chebyshev scalarization function (Wierzbicki, 1982), denoted as $s_{\boldsymbol{\lambda}}(\tilde{f}_\ell(x)) = -\max_\ell[\lambda_\ell|\tilde{f}_\ell(x) - z_\ell^*|] - \gamma \sum_{i=\ell}^{L} |\tilde{f}_\ell(x) - z_\ell^*|$, where $z^*$ is the ideal or utopian vector, $L$ is the number of objective dimensions, and $\gamma$ is a constant weighing the linear term being added to the traditional Chebyshev scalarization function (Chugh, 2019). We find that this scalarization function performs better since the Chebyshev component allows for non-convex Pareto frontier sampling and the augmented term assists with sampling within the hard bound.

**Step 2: Pareto Frontier Sparsification.** All of the baseline algorithms were run immediately after step (1) is completed. In order to ensure all of the algorithms sample an equal number of points, we first run MoSH-Sparse before deciding for greedy and random baseline algorithms to sample the same number of points.

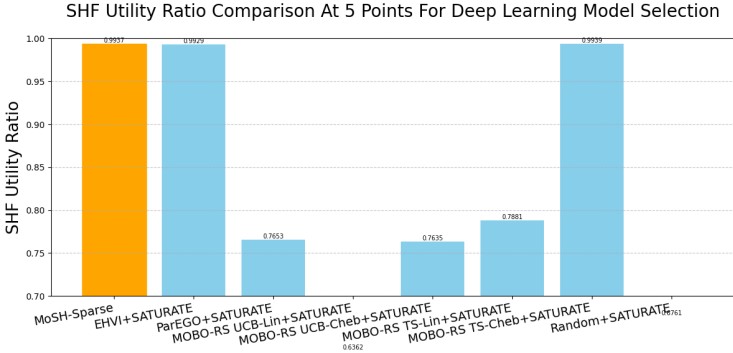

Figure 19: Bar plot illustrating the SHF utility ratio obtained by our method, MoSH-Sparse, compared to the other baselines on the deep learning model selection application. Only the dense set, from step 1, changes for each bar.

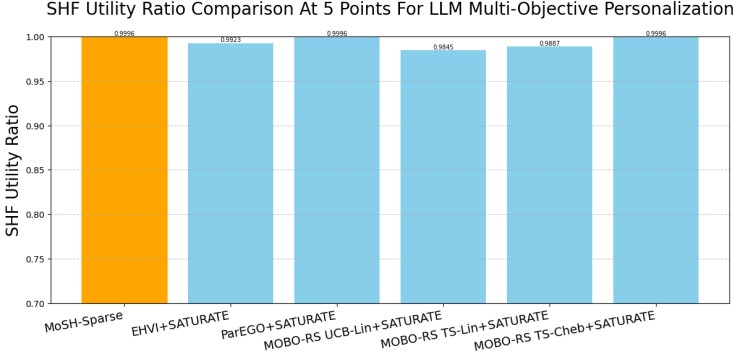

Figure 20: Bar plot illustrating the SHF utility ratio obtained by our method, MoSH-Sparse, compared to the other baselines on the LLM personalization problem. Only the dense set, from step 1, changes for each bar.

### A.5.2 PERFORMANCE EVALUATION

To calculate the soft-hard fill distance, we first computed a grid search of points on the Pareto frontier, offline, for each experiment. For each experiment, the soft-hard fill distance was then taken with respect to that computed set. As a result, the results for the soft-hard fill distance are somewhat dependent on this offline set of points – however, we ensured that it was kept constant for each set of experiments. In some cases, notably in Figure 4, the soft-hard fill distance does not monotonically decrease. This is due to the heuristic we use in cases where there are no points which have been sampled within either the soft or hard regions, at some iteration. In such a case, we select the worst-case point from set $D$ to represent set $C$ and calculate the metric using that. Once there exists points sampled within the soft or hard regions, we remove that worst-case point and instead use the sampled points. Since the sampled points may result in a soft-hard fill distance value worse than the worst-case point from $D$, the metric may increase – although the general trend will remain the same. For $\kappa$ in the soft region distance-weighted score, we used the Euclidean distance.

