# OpenReview forum: "MoSH: Modeling Multi-Objective Tradeoffs with Soft and Hard Bounds"
_ICLR.cc/2025/Conference — Submitted to ICLR 2025_

### Official Review · Reviewer_wJhw · 2024-10-24

**Soundness:** 2
**Presentation:** 2
**Contribution:** 2
**Rating:** 3
**Confidence:** 3

**Summary:**

The authors proposed a two-step framework for MOO that incorporates user defined soft and hard bounds on objectives and generates a compact set of Pareto optimal solutions for human decision makers to review. The framework consists of

1. MoSH-Dense: a Bayesian optimization approach using random scalarizations and soft-hard functions to densely sample the Pareto frontier.
2. MoSH-Sparse: a submodular optimization method that is robust to unknown human preference, to select a small, diverse subset of solutions from the dense set.

The authors evaluated this framework on a set of synthetic problems and real-world applications, including brachytherapy treatment planning, and showed merit of this method.

**Strengths:**

Originality:
1. This approach creatively incorporates the concepts of hard constraints and soft preferences into BO formulation, providing a more nuanced way to express decision-maker preferences.
2. The two-step framework is more user friendly: through sparcification of the originally found Pareto optimal solutions, it is easier for human DMs to pick the solutions to their preferences. Therefore, I would image there is fair amount of practical usage of this approach.

Quality:
1. This paper provides comprehensive theoretical analysis of the MoSH-Sparse algorithm, with provable guarantees on the near-optimality of the selected sparse set.

Clarity:
1. I think this paper is pretty well structured, and the proposed method is clearly presented.

Significance:
1. MoSH addresses a practical challenge in MOO by providing a way to generate a small but diverse set of solutions that respect user defined constraints. This has great potential impact to MOO applications.

**Weaknesses:**

1. The algorithmic novelty is fairly limited: for the dense sampling procedure, the primary modification is the incorporation of SHFs into the acquisition function, while the rest largely follow existing techniques from Paria et al (2019). The sparcification procedure follows standard active learning approach. While the problem formulation is novel and interesting, I do not think there is enough methodological innovation to meet the high standard of ICLR.

2. The paper lacks discussion of design choices of SHFs given it is quite important in the proposed method. Specifically, is it the case that decision-makers are always able to meaningfully specify soft and hard bounds? What if the bounds are uncertain or change over the exploration of the experiments? Any alternative utility function classes other than the piecewise-linear SHF discussed in this paper, and is the overall method robust to the choice of SHFs?

**Questions:**

1. The piecewise linear SHF seems somewhat arbitrary. Have you explored other functional forms, and what motivated this particular choice? The reasons need to be justified.

2. For the brachytherapy application, please add discussion about how the clinical experts use this system and their feedback. Additionally, compared with other MOO interactive solutions mentioned in this paper’s related works section, what is the advantage of using the proposed method here instead of those interactive mechanisms? The experiment section lacks comparisons against those interactive MOO methods.

---

> ### Author Response · Authors · 2024-11-21
> **Author Response 1/2**
>
> Dear Reviewer wJhw,
>
> Thank you very much for your thoughtful review and valuable feedback. We really appreciate it!
>
> We are glad that you find the two-step framework to be very user-friendly. We also believe there is large practical interest in using soft and hard bounds for real-world problems. Furthermore, we appreciate your support in the overall clarity and significance of this proposed method.
>
> We wanted to respond to and clarify some details you mentioned:
>
> >*The algorithmic novelty is fairly limited: for the dense sampling procedure, the primary modification is the incorporation of SHFs into the acquisition function, while the rest largely follow existing techniques from Paria et al (2019). The sparcification procedure follows standard active learning approach. While the problem formulation is novel and interesting, I do not think there is enough methodological innovation to meet the high standard of ICLR.*
>
> We would like to emphasize a few points:
>
> - **Our primary contribution is the novel conceptual framework of soft-hard bounds for multi-objective optimization (formalization of the problem, our proposed method, soft-hard evaluation metrics, and broad set of experiments (across cancer treatment, LLM personalization, and more), all native to this novel setting).** To the best of our knowledge, this setting has not been studied before in the multi-objective optimization setting, yet is ubiquitous in many real-world problems. As you had also mentioned, we believe that this general setting of allowing for decision makers to use soft and hard bounds when imparting their expertise is potentially of great practical utility. We hope that this may serve as a basis for and pave the way for future works in this important area of multi-objective optimization.
>
> - We are the first to formalize this problem with a simple two-step process which returns an easily interpretable, small set of points to the user according to the soft-hard bounds they had specified.
>
> - As part of this novel conceptual framework, we also **introduced novel evaluation metrics which are native to this soft-hard setting – rather than reusing existing ones. We additionally rigorously evaluated our setting, both theoretically and empirically, while also introducing a novel experiment on multi-objective personalization for LLMs (which had previously not been evaluated on in the multi-objective optimization literature, but shows our setting’s broad relevance).** (Section 5.3.2 of the paper)
>
> >*The paper lacks discussion of design choices of SHFs given it is quite important in the proposed method. Specifically, is it the case that decision-makers are always able to meaningfully specify soft and hard bounds? What if the bounds are uncertain or change over the exploration of the experiments? Any alternative utility function classes other than the piecewise-linear SHF discussed in this paper, and is the overall method robust to the choice of SHFs?*
>
> >*The piecewise linear SHF seems somewhat arbitrary. Have you explored other functional forms, and what motivated this particular choice? The reasons need to be justified.*
>
> - Thank you very much for raising this interesting point. We would first like to emphasize that our **main focus of this paper is sampling points on the Pareto frontier according to some set of soft and hard bounds; in other words, we did not intend to cover any of the interactive aspects of the soft and hard bounds (which we do find very interesting, however)**. For our paper, we assume that the decision-maker is able to meaningfully specify some set of soft and hard bounds. In a real-world setting, we would expect for the bounds to change over the exploration of experiments – however, as we had mentioned, this is out of scope of our paper.
> - We primarily focused on **piecewise-linear utility function SHFs due to their simplicity**. However, we believe any function class which satisfies our soft-hard setting assumptions, described in Section 2.2 of the paper, will suffice. We have updated the writing to make this more clear.

---

> ### Author Response · Authors · 2024-11-21
> **Author Response 2/2**
>
> >*For the brachytherapy application, please add discussion about how the clinical experts use this system and their feedback. Additionally, compared with other MOO interactive solutions mentioned in this paper’s related works section, what is the advantage of using the proposed method here instead of those interactive mechanisms? The experiment section lacks comparisons against those interactive MOO methods.*
>
> Thank you for the question.
> - We highlighted this briefly in the second paragraph of the Introduction, Section 1. As additional context: through discussions with clinicians, we found that they have internal preferences, often in the form of soft and hard constraints, which they found difficult to express to traditional healthcare systems. These come in forms, such as “We would definitely like for the cancer tumor to have 90% radiation dosage coverage, however, it would be more preferable if the cancer tumor could reach 95% radiation dosage coverage” – similarly for the radiation dosage to healthy organs nearby, although with the aim to minimize dosage there.
> - Prior to our proposed SHFs, there were no other methods which directly and intuitively corresponded to such clinical feedback. We further showed that this type of feedback is relevant for countless other real-world applications. Specifically, in our paper we conducted experiments across five different domains. We recently added into the writing an additional experiment on multi-objective personalization of large language models to showcase further its potential widespread applicability.
> - Compared with other interactive methods in the literature, the advantage of our method is that it is **easily interpretable to the practitioner and directly encodes the practitioner’s feedback into finding a small set of points on the proper region of the Pareto frontier – which did not exist in prior methods, to the best of our knowledge**. As we also mentioned above, the interactive aspect is out of scope of our paper (but we do find it very interesting).
>
> Again, thank you very much for your feedback. We hope we've addressed your concerns. Please let us know if there are any additional points to discuss.

---

> ### Author Response · Authors · 2024-11-23
> **Author Follow-Up**
>
> Dear Reviewer wjhw, since the discussion period is ending in a few days (Nov. 26), we would love to hear if our response has addressed your concerns. Thank you and please let us know if you have any other questions!

---

### Official Review · Reviewer_qibo · 2024-10-26

**Soundness:** 3
**Presentation:** 2
**Contribution:** 3
**Rating:** 5
**Confidence:** 3

**Summary:**

This paper considers when performing multi-objective optimization, for each objective $f(x)$ there is a soft lower bound $\alpha_{lS}$ that is ideally satisfied and a hard lower bound $\alpha_{lH}$   that must be satisfied. Additionally, it proposes a max-min framework for scalarization-based multi-objective optimization, aiming to find solutions that are robust against variations in the scalarization weights $\lambda$ within a bounded set $\Lambda$ to enhance solution diversity.

The paper first proposes to leverage a pice-wise linear scalarization function to incorporate the soft and hard objective constraints. For the max-min formulation of optimization, given the computational challenges of directly solving the max-min problem, the paper introduces a two-step approximation process. The first step involves solving a max-average problem to generate a set of promising candidate solutions. The second step refines this set by removing points that do not significantly affect the worst-case utility ratio, leveraging the submodular properties of the utility function to apply the Robust Submodular Observation Selection (RSOS) framework and the submodular saturation algorithm.

The proposed algorithm is empirically tested on synthetic and real-world problems, demonstrating its practical applicability.

**Strengths:**

- The paper presents an interesting problem of multi-objective optimization considering soft and hard lower bounds.
- Perhaps the more interesting thing is the max-min framework to achieve robustness against user preference weights is novel, which might provide a new perspective of enhancing diversity in Pareto frontier search.
- The practical approximation using a two-step process and leveraging submodular optimization techniques is sound, as it offers a feasible solution to an otherwise intractable problem.

**Weaknesses:**

- Theoretical Analysis: The main drawback of this paper is, though heavily constructed on relaxations/approximations, does not provide adequate theoretical insights about the method itself, to elaborate:
    - The transformation from the max-min to the max-average problem is presented as a practical approximation, but the paper lacks theoretical analysis regarding its impact on the optimization results. Also, it is expected to be increasingly difficult for the expectation to be conducted when the number of objectives is increasing.
    - Similarly, which might also be because of a writing issue, the relaxation to a decomposable form to apply the RSOS framework is sound but without any theoretical insight.
     - If such analysis is non-trivial, a more pragmatic workaround could be to use some grid (or Monte Carlo) based approximation of the original min in max-min framework, such that it provides a good reference of the ground truth behavior of the original framework, but seems has not been done in the paper.

- Clarity:
    - All the plot quality is rather poor; please increase the size of all the image's legends and label and title font size. Also please consider avoiding snake case name style (e.g., consider avoiding using MOBO_RS_UCB_Lin)

**Questions:**

- It seems in experiments, it is compared with other acquisition functions with step 1 only (Sec 323), I am puzzled by such a motivation here, cause this is essentially comparing max-average with other MOBO acq instead of max-min (which is originally thought to be approximated by the combination of two steps)
- The objective constraints and the scalarization weights $\mathbf{\lambda}$ are handled almost independently, however, it is natural that they should share some dependence, why this is not considered.
- The explainability of the utility function definition: The utility function definition is more complicated than necessary at least when originally read Eq. 1, especially because of the not well-motivated saturation points, why this is needed here, is that because of the suitableness of the submodular saturation algorithm? please state it clearly after Eq. 1.

---

> ### Author Response · Authors · 2024-11-21
> **Author Response 1/2**
>
> Dear Reviewer qibo,
>
> Thank you very much for your thoughtful review and valuable feedback. We really appreciate it!
>
> We are glad that you find the multi-objective optimization problem considering soft and hard lower bounds to be very interesting. We also very much appreciate your support of the max-min framework and two-step process which we propose.
>
> We wanted to respond and clarify some details you mentioned:
>
> >*Theoretical Analysis: The main drawback of this paper is, though heavily constructed on relaxations/approximations, does not provide adequate theoretical insights about the method itself, to elaborate:
> The transformation from the max-min to the max-average problem is presented as a practical approximation, but the paper lacks theoretical analysis regarding its impact on the optimization results. Also, it is expected to be increasingly difficult for the expectation to be conducted when the number of objectives is increasing.
> Similarly, which might also be because of a writing issue, the relaxation to a decomposable form to apply the RSOS framework is sound but without any theoretical insight.
> If such analysis is non-trivial, a more pragmatic workaround could be to use some grid (or Monte Carlo) based approximation of the original min in max-min framework, such that it provides a good reference of the ground truth behavior of the original framework, but seems has not been done in the paper.*
>
> Thank you for your comments about this.
> - In short, **we conducted additional experiments based on your suggestion and validated the computational difficulty reason for converting from the max-min to the max-average problem** (Section A.1.1 of the paper).
> - In terms of the computational results, we ablated across multiple step sizes for the discretization of the input and lambda space and evaluated a greedy-based algorithm on all settings (as you had suggested). Specifically, $\textit{discrete-greedy-0.05}$ (where the last value indicates the grid discretization step size) and $\textit{discrete-greedy-0.10}$ take, on average, **20x and 3x longer (in seconds) than our proposed method MoSH-Dense, respectively, to select the next sample $x_t$ at iteration $t$**. This computational disparity is further exacerbated as the size of the input space dimensionality increases. However, as expected, as $\delta \rightarrow 0$, the metrics improve and move closer to those of MoSH-Dense. Finally, we note that, as described in Section 4, the greedy algorithm may perform arbitrarily bad when solving the max-min problem. We leave for future work exploration of other algorithms which are designed for our setting described in Section 3; notably, access to some noisy and expensive black-box function. Further details have been added in the writing (Appendix 1.1).
> - We are currently running more experiments for the other domains and will add them into the paper.
>
> - Regarding your comment about the decomposable form for the RSOS framework, we are assuming that you are referring to Equation 4 in Section 4 of the writing. If so, **we do not view this as a relaxation but more as a slight reframing of the problem**, which we notice corresponds to works in the robust submodular observation selection literature. One of the assumptions of the robust submodular observation selection literature is that the inner objective term, $F_\lambda$, is submodular. We prove that in Appendix A.3.
> - On the other hand, if you are referring to the relaxation component of the SATURATE algorithm, that is further explained in Krause et al., 2008. In short, the relaxation part of the SATURATE algorithm allows for theoretical guarantees on the optimality of the algorithm, in terms of covering the input set of points, albeit at a slightly higher cost (i.e. selecting potentially more samples). Please let us know if this is not clear.
>
> - To provide additional theoretical insights into our method, **we added into the writing a theorem which shows a lower bound for the SHF utility ratio as $t \rightarrow \infty$**. Overall, we show that as $t \rightarrow \infty$, the SHF utility ratio approaches 1 (Section A.3 of the paper).
> This intuitively proves that our proposed MoSH-Sparse is able to offer sufficiently dense coverage, as intended.
> - To prove the theorem, we leveraged the Lipschitz continuity of our proposed soft-hard bound utility function as well as a regret decomposition form from Russo and Van Roy (2014) and Paria et al. (2020) (details in Appendix).
>
> >*All the plot quality is rather poor; please increase the size of all the image's legends and label and title font size. Also please consider avoiding snake case name style (e.g., consider avoiding using MOBO_RS_UCB_Lin)*
>
> Thank you very much for your constructive criticism. **We have updated the writing with higher-quality plots — increased text/label size and removal of snake case names.**

---

> ### Author Response · Authors · 2024-11-21
> **Author Response 2/2**
>
> >*It seems in experiments, it is compared with other acquisition functions with step 1 only (Sec 323), I am puzzled by such a motivation here, cause this is essentially comparing max-average with other MOBO acq instead of max-min (which is originally thought to be approximated by the combination of two steps)*
>
> For our experiments evaluating step 1, we wished to compare our proposed method (MoSH-Dense) with others in terms of how well they each can densely sample the soft-hard-bound defined regions of the Pareto frontier. If we are understanding correctly, you are asking why we, for our step 1 evaluation, do not compare the max-average formulation against the max-min formulation (similar to your comment regarding the theoretical analysis component above). Our motivation for not initially having such a comparison is because we did not see it as a valid baseline due to the computational cost and need for discretization (making it impractical for real-world scenarios). However, based on your suggestions, we did perform the experiment to observe the differences (as mentioned above). We thank you for bringing this up as we believe it further strengthens our paper.
>
> >*The objective constraints and the scalarization weights $\lambda$ are handled almost independently, however, it is natural that they should share some dependence, why this is not considered.*
>
> Thank you for this question. The SHFs, for the objective constraints, are handled independently from the scalarization weights $\lambda$ because we find that doing so allows for more flexibility by providing a more interpretable and practical feedback mechanism while still allowing for our proposed method to sample diversely across the soft-hard-bound specified region of the Pareto frontier.
>
> >*The explainability of the utility function definition: The utility function definition is more complicated than necessary at least when originally read Eq. 1, especially because of the not well-motivated saturation points, why this is needed here, is that because of the suitableness of the submodular saturation algorithm? please state it clearly after Eq. 1.*
>
> Thank you for bringing this up. The saturation point is there to prevent “exploding utility values” for points on the Pareto frontier which contain values which satisfy the soft constraint by a large amount. Intuitively, this would place unnecessarily high weight on such points, even if they possess values in the other objectives which are non-ideal.
>
> From a practical perspective, say, with the brachytherapy clinical scenario, a treatment plan which satisfies the soft constraint by a large amount for one dimension will often exhibit harmful values in the other dimensions. We aimed to reflect that in the SHF. The writing has been updated to reflect this, below Equation 1.
>
> Again, thank you very much for your feedback. We hope we've addressed your concerns. Please let us know if there are any additional points to discuss.

---

> ### Author Response · Authors · 2024-11-23
> **Author Follow-Up**
>
> Dear Reviewer qibo, since the discussion period is ending in a few days (Nov. 26), we would love to hear if our response has addressed your concerns. Thank you and please let us know if you have any other questions!

---

### Official Review · Reviewer_vZPb · 2024-11-04

**Soundness:** 2
**Presentation:** 2
**Contribution:** 2
**Rating:** 3
**Confidence:** 5

**Summary:**

The authors consider sampling the Pareto frontier of a multi-objective optimization problem. They first sample the Pareto frontier by random scalarizations (which does not seem to come with any guarantees, at least for non-convex problems). Subsequently, they "sparsify" the sample. They show that for some preferences lambda, the latter problem is submodular and propose to use an algorithm of Krause for robust submodular observation selection (JMLR, 2008).

**Strengths:**

The problem is of considerable practical interest.

**Weaknesses:**

The paper is rather confusingly written. While the method does not come with any guarantees overall, the fact may be lost on the casual reader.

The empirical results are not convincing. Specifically, the comparison does not utilize any standard benchmarks, such as the one from CEC 2009 (Zhang et al. 2008).

The comparison does not extend to any methods with overall guarantees, e.g., by Christos Papadimitriou and Mihalis Yannakakis (e.g., https://dl.acm.org/doi/10.1145/375551.375560, https://link.springer.com/chapter/10.1007/3-540-44634-6_1).

The "Related work" discussion is rather biased and methods with overall guarantees are not even cited.

**Questions:**

How would your performance compare to the performance of the methods of Christos Papadimitriou and Mihalis Yannakakis (e.g., https://dl.acm.org/doi/10.1145/375551.375560, https://link.springer.com/chapter/10.1007/3-540-44634-6_1), or at least to the guarantees they have?

---

> ### Author Response · Authors · 2024-11-21
> **Author Response 1/2**
>
> Dear Reviewer vZPb,
>
> Thank you very much for your review and feedback. We really appreciate it!
>
> We are glad that you agree about the practical interest of the multi-objective optimization problem. We further believe our proposed method has potential for great practical usage in many applications, several of which we highlighted in the paper (and also mentioned by Reviewer wJhw).
>
> We wanted to respond and clarify some details you mentioned:
>
> >*The paper is rather confusingly written. While the method does not come with any guarantees overall, the fact may be lost on the casual reader.*
>
> Thank you for pointing this out.
> - Based on your suggestion, **we have added into the writing an additional theorem proving a lower bound for Step 1 of our algorithm**. In doing so, we show that Step 1 of our algorithm achieves an SHF Utility Ratio of 1 as $t \rightarrow \infty$ (Section A.3 of the paper).
> - To prove the theorem, we leveraged the Lipschitz continuity of our proposed soft-hard bound utility function as well as a regret decomposition form from Russo and Van Roy (2014) and Paria et al. (2020). The proof is in the Appendix and referenced in the main text. Thank you again for this suggestion as we believe it greatly improves our paper.
> - **We updated the writing to hopefully be more clear. Specifically, we also clarified our theoretical guarantees for each of the two steps.**
>
> >*The empirical results are not convincing. Specifically, the comparison does not utilize any standard benchmarks, such as the one from CEC 2009 (Zhang et al. 2008).*
>
> Thank you for your constructive criticism. We believe that some of the experiments we conducted already use standard benchmarks in the multi-objective decision making/machine learning literature, notably Branin-Currin and FourBarTruss.
> - Branin-Currin has been used in Daulton et al. (2020), Zhao et al. (2022), Rashidi et al. (2024) and more.
> - FourBarTruss, or generally, problems from the REPROBLEM dataset (Tanabe et al. (2020)), have been used in Daulton et al. (2020), Ament et al. (2023), Lin et al. (2022), and more.
> - Hyperparameter optimization for neural networks is a fairly standard problem commonly evaluated in the literature as well. We followed a similar setup to the following papers: Hernandez-Lobato et al. (2016) and Abdolshah et al. (2019).
> - After taking a look at the multi-objective test instances from CEC 2009 (Zhang et al. 2008), we do not believe the goals for which they were designed matches our setting of **soft-hard-bound-defined Pareto frontiers**. Furthermore, we still feel that the test instances we conducted our experiments on are more standard and representative of real-world problems. Notably, FourBarTruss is from the REPROBLEM (Tanabe et al, 2020) test suite, which is explicitly modeled after real-world problems.
> - **To help with making the empirical results more convincing, and based on your suggestion, we have added into the writing a novel experiment regarding multi-objective personalization of large language models.** For the LLM experiment, we use soft-hard bounds on two competing objectives - conciseness and informativeness - and demonstrate our method's abilities to consistently sample output responses within such soft and hard bounds (Section 5.3.2 of the paper). In doing so, we demonstrate further the broad applicability of our method.
>
> References:
> - Samuel Daulton, Maximilian Balandat, and Eytan Bakshy. "Differentiable Expected Hypervolume Improvement for Parallel Multi-Objective Bayesian Optimization". NeurIPS, 2020.
> - Yiyang Zhao, Linnan Wang, Kevin Yang, Tianjun Zhang, Tian Guo, and Yuandong Tian. "Multi-Objective Optimization By Learning Space Partitions". ICLR, 2022.
> - Bahador Rashidi, Kerrick Johnstonbaugh, and Chao Gao. "Cylindrical Thompson Sampling for High-Dimensional Bayesian Optimization". AISTATS, 2024.
> - Sebastian Ament, Samuel Daulton, David Eriksson, Maximilian Balandat, and Eytan Bakshy. "Unexpected Improvements to Expected Improvement for Bayesian Optimization". NeurIPS, 2023.
> - Xi Lin, Zhiyuan Yang, Xiaoyuan Zhang, and Qingfu Zhang. "Pareto Set Learning for Expensive Multi-Objective Optimization". NeurIPS, 2022.
> - Daniel Hernandez-Lobato, Jose Miguel Hernandez-Lobato, Amar Shah, and Ryan P. Adams. "Predictive Entropy Search for Multi-objective Bayesian Optimization". ICML, 2016.
> - Majid Abdolshah, Alistair Shilton, Santu Rana, Sunil Gupta, Svetha Venkatesh. "Multi-objective Bayesian optimisation with preferences over objectives". NeurIPS, 2019.
> - Ryoji Tanabe and Hisao Ishibuchi. "An easy-to-use real-world multi-objective optimization problem suite". Applied Soft Computing, 2020.

---

> ### Author Response · Authors · 2024-11-21
> **Author Response 2/2**
>
> >*The comparison does not extend to any methods with overall guarantees, e.g., by Christos Papadimitriou and Mihalis Yannakakis (e.g., https://dl.acm.org/doi/10.1145/375551.375560, https://link.springer.com/chapter/10.1007/3-540-44634-6_1).*
>
> >*How would your performance compare to the performance of the methods of Christos Papadimitriou and Mihalis Yannakakis (e.g., https://dl.acm.org/doi/10.1145/375551.375560, https://link.springer.com/chapter/10.1007/3-540-44634-6_1), or at least to the guarantees they have?*
>
> Thank you for sharing the references.
> - However, after reading through them, we do not feel they are directly comparable. The papers you shared, both from 2001, consider vastly different settings and are specific to the database query optimization problem with two objectives - cost and delay. If, however, you are suggesting additional theoretical guarantees, we have added into the writing an additional theorem with the lower bound for the dense sampling step (MoSH-Dense) of our algorithm.
> - To emphasize the main contribution of our work and how it differs greatly from the references you shared: **our primary contribution is the novel conceptual framework of soft-hard bounds for multi-objective optimization (formalization of the problem, our proposed method, soft-hard evaluation metrics, and broad set of experiments (across cancer treatment, LLM personalization, and more), all native to this novel setting). We emphasize that our primary interest is only a subset of the Pareto frontier as defined by such soft-hard bounds.**
>
> >*The "Related work" discussion is rather biased and methods with overall guarantees are not even cited.*
>
> It would be very helpful if you could please point us to any papers with overall guarantees we are missing. As of now, the related works section we have contains multiple papers with overall guarantees, including Paria et al. (2020), and Zuluaga et al. (2016).
>
> Again, thank you very much for your feedback. We hope we've addressed your concerns. Please let us know if there are any additional points to discuss.

---

> > ### Comment · Reviewer_vZPb · 2024-12-01
> > **Thank you**
> >
> > Dear all,
> >
> > many thanks for the careful rebuttal and the additional results in the appendix.
> >
> > > It would be very helpful if you could please point us to any papers with overall guarantees we are missing.
> >
> > I meant the results of Christos Papadimitriou and Mihalis Yannakakis (e.g., https://dl.acm.org/doi/10.1145/375551.375560, https://link.springer.com/chapter/10.1007/3-540-44634-6_1), which I have mentioned in the original review. They provide guarantees as to the sampling of the Pareto front, which combines both papers of yours.
> >
> > I really do think that this is the best "point of reference", in terms of the problem. Also, these papers are not recent, little-known preprints. Indeed, they are classics in the field and Christos Papadimitriou (https://scholar.google.co.uk/citations?user=rXYLXJMAAAAJ&hl=en&oi=ao) is one of the most prominent theoretical computer scientists alive, so it could be expected that you would know them and explain how you improve upon them.

---

> ### Author Response · Authors · 2024-11-23
> **Author Follow-Up**
>
> Dear Reviewer vZPb, since the discussion period is ending in a few days (Nov. 26), we would love to hear if our response has addressed your concerns. Thank you and please let us know if you have any other questions!

---

> ### Author Response · Authors · 2024-12-02
> **Response From Authors**
>
> Dear Reviewer vZPb,
>
> Thank you very much for your response.
>
> We have carefully examined the references you shared, and while we greatly respect these works and acknowledge their significant contributions to the field, we believe they address a fundamentally different problem than the one we tackle in our paper.
>
> Specifically, Papadimitriou and Yannakakis (2001) consider the Mariposa Coordinator Problem in database query optimization. Their work assumes access to a discrete set of query plans and adapts existing greedy/dynamic programming algorithms. They provide valuable proofs regarding:
> - the Mariposa’s algorithm in finding the full convex Pareto curve of cost-delay tradeoffs
> - a general dynamic programming algorithm for computing the full $\epsilon$-Pareto curve, which is a polynomially small set of solutions
> - a pseudo-polynomial dynamic programming-based algorithm for computing the full $\epsilon$-Pareto curve for the more general problem where the strides are not predetermined
>
> **Our work takes a different direction that we believe complements rather than overlaps with their contribution.** Our approach differs in several key aspects:
> - We operate in a continuous rather than discrete input space, and we focus on identifying an easily navigable set of Pareto-optimal points on a *user-specified subset* of the Pareto frontier, rather than the entire frontier as in Papadimitriou and Yannakakis (2001)
> - We show that our easily navigable set of Pareto-optimal points is theoretically guaranteed to optimally cover the original, dense set of Pareto-optimal points, with a slightly higher cost
> - Our subset of the Pareto frontier is *specified by the user with soft and hard bounds* (not considered in Papadimitriou and Yannakakis (2001)). We coin the term soft-hard functions (or SHFs) which we use to compute the utility values of different regions in the subset of the Pareto frontier
> - Similar to Papadimitriou and Yannakakis (2001), *we provide theoretical guarantees* for our algorithms. In our case, we provide theoretical guarantees for both components of our proposed two-step algorithm (one of which was added during the discussion period, as previously described)
>
> All the details that we discussed above are presented in our paper, initially highlighted on the second page in the latter part of the Introduction section. We have also added some clarifying details during the discussion stage, which we hope helps to make our contribution even clearer. **If you have any specific suggestions regarding how we can further clarify our contributions/differences in the writing, please let us know.**
>
> While we deeply appreciate you bringing these references to our attention, we respectfully suggest that the differences in problem formulation, approach, and objectives make our contribution distinct and complementary to the existing literature you've cited.
>
> Thank you again for your feedback and engagement with our work. We remain open to further discussion and welcome any additional questions or concerns you may have.

---

### Official Review · Reviewer_mTMT · 2024-11-07

**Soundness:** 2
**Presentation:** 2
**Contribution:** 2
**Rating:** 5
**Confidence:** 3

**Summary:**

The paper proposes incorporating real-world constraints into multi-objective optimization tasks by converting the reward function into a newly introduced soft-hard function (SHF). Existing Multi-Objective Bayesian Optimization algorithms and Greedy Submodular Partial Cover algorithms are then employed to achieve sparse coverage of the Pareto frontier with respect to the introduced SHF utility ratio.

**Strengths:**

a. The algorithm is motivated by real-world applications, with a careful illustration of the objective design.

b. The sample efficiency and sparsification efficiency are justified with theoretical analysis and empirical results on various tasks.

**Weaknesses:**

a. Questionable Necessity of SHF Utility Ratio: The necessity of introducing the SHF utility ratio is debatable. The algorithm essentially aims to offer sparse coverage of the Pareto frontier. Existing work by Zuluaga et al. (2016), as mentioned by the authors, focuses on sample-efficient identification of the $\epsilon$-accurate Pareto set. The notion of an $\epsilon$-accurate Pareto set is a more principled and interpretable objective for sample-efficient algorithms. I respectfully disagree with the authors' claim that it resembles hypervolume-based heuristics.

b. Limited Novelty of the Proposed Algorithm: The novelty of the proposed algorithm is arguable. The algorithm essentially converts the multi-objective optimization objective using the introduced SHF utility function to incorporate constraints, and the SHF utility ratio measures the coverage of the SHF Pareto frontiers. Beyond these introduced notions, components like random scalarization, the Bayesian Optimization acquisition procedure, and submodular optimizations, along with their theoretical results, are mostly off-the-shelf. Additionally, the use of existing methods—especially the MOBO algorithm by Paria et al. (2019) and the GPC algorithm by Krause et al. (2008)—is not clearly stated in the main paper but deferred to the appendix.

c. Clarity and Readability Issues: There are clarity and readability issues in the current presentation. Key components of the algorithm, including Algorithm 3 and the acquisition function, are deferred to the appendix, which hinders understanding. Furthermore, the figures, especially those in Section 5, need to be polished for better readability.

d. Classification of Related Work: A minor point—another reference by Malkomes et al. (2021) is arguably classified as a Level Set Estimation (LSE) method. From my perspective, it also addresses identifying sparse, diversified coverage of constrained optimal candidates.

***References***
- Zuluaga, Marcela, and Andreas Krause. "e-pal: An active learning approach to the multi-objective optimization problem." Journal of Machine Learning Research 17, no. 104 (2016): 1-32.
- Malkomes, Gustavo, Bolong Cheng, Eric H. Lee, and Mike Mccourt. "Beyond the pareto efficient frontier: Constraint active search for multiobjective experimental design." In International Conference on Machine Learning, pp. 7423-7434. PMLR, 2021.

**Questions:**

A follow-up work by Zhang and Golovin (2020) builds on Paria et al. (2019) and could allow for a more consistent analysis in terms of optimization performance, rather than assuming Algorithm 1 offers sufficiently dense coverage. The authors might consider incorporating methods from Zhang and Golovin (2020) to improve the completeness of their analysis.

***Reference***

Zhang, Richard, and Daniel Golovin. "Random Hypervolume Scalarizations for Provable Multi-Objective Black Box Optimization." In International Conference on Machine Learning, pp. 11096-11105. PMLR, 2020.

---

> ### Author Response · Authors · 2024-11-21
> **Author Response 1/2**
>
> Dear Reviewer mTMT,
>
> Thank you very much for your thoughtful review and valuable feedback. We really appreciate it!
>
> We are glad that you agree that one of the strengths of the algorithm is how it’s motivated by real-world applications. We believe it has potential for great practical usage in many applications, several of which we highlighted in the paper.
>
> We wanted to respond and clarify some details you mentioned:
>
> > *Questionable Necessity of SHF Utility Ratio: The necessity of introducing the SHF utility ratio is debatable. The algorithm essentially aims to offer sparse coverage of the Pareto frontier. Existing work by Zuluaga et al. (2016), as mentioned by the authors, focuses on sample-efficient identification of the $\epsilon$-accurate Pareto set. The notion of an $\epsilon$-accurate Pareto set is a more principled and interpretable objective for sample-efficient algorithms. I respectfully disagree with the authors' claim that it resembles hypervolume-based heuristics.*
>
> - The primary purpose for explicitly defining the SHF Utility Ratio, which we do at the bottom of Section 2.3, is to easily refer back to it when evaluating the utility obtained by the decision-maker. We use it for each of the five applications we evaluated on. Intuitively, it provides us with a notion for measuring the soft-hard-bound defined utility obtained by the decision-maker. We have updated the writing to make this more clear.
>
> - Regarding Zuluaga et al. (2016), we thank you for pointing that out. While we believe Zuluaga et al. (2016) is a valuable contribution, we would like to emphasize that our proposed method is different in that **we only focus on a subset of the Pareto frontier, as defined by soft and hard constraints**. This allows for a more intuitive setting which many practitioners may find useful, as also mentioned by Reviewer wJhw. We have clarified the writing around that in Section 6.
>
> > *Limited Novelty of the Proposed Algorithm: The novelty of the proposed algorithm is arguable. The algorithm essentially converts the multi-objective optimization objective using the introduced SHF utility function to incorporate constraints, and the SHF utility ratio measures the coverage of the SHF Pareto frontiers. Beyond these introduced notions, components like random scalarization, the Bayesian Optimization acquisition procedure, and submodular optimizations, along with their theoretical results, are mostly off-the-shelf. Additionally, the use of existing methods—especially the MOBO algorithm by Paria et al. (2019) and the GPC algorithm by Krause et al. (2008)—is not clearly stated in the main paper but deferred to the appendix.*
>
> Thank you for pointing this out. We would like to emphasize a couple of points:
>
> - **Our primary contribution is the novel conceptual framework of soft-hard bounds for multi-objective optimization (formalization of the problem, our proposed method, soft-hard evaluation metrics, and broad set of experiments (across cancer treatment, LLM personalization, and more), all native to this novel setting).** We believe that our general setting of allowing for decision makers to use soft and hard bounds when imparting their expertise is novel and potentially of great practical utility (as also mentioned by Reviewer wJhw). We hope that this may serve as a basis for and pave the way for future works in this important area of multi-objective optimization.
>
> - Based on your suggestion, **we updated the writing to make our novel contributions more clear, performed another set of experiments for the novel multi-objective setting of personalization for LLMs (Section 5.3.2 of the paper), and added theoretical analyses to further support our proposed method (highlighted in response 2/2) (Section A.3 of the paper)**. For the LLM experiment, we use soft-hard bounds on two competing objectives - conciseness and informativeness - and demonstrate our method's abilities to consistently sample output responses within such soft and hard bounds. In doing so, we demonstrate further the broad applicability of our method.
>
> - Furthermore, to our knowledge, there are no other existing works which incorporate both soft and hard bounds into the multi-objective optimization process. Although we leverage existing works for steps 1 (MoSH-Dense) and 2 (MoSH-Sparse) of our proposed method, we find that they do well in achieving our objective and rigorously support them through theory and experiments.

---

> ### Author Response · Authors · 2024-11-21
> **Author Response 2/2**
>
> > *Clarity and Readability Issues: There are clarity and readability issues in the current presentation. Key components of the algorithm, including Algorithm 3 and the acquisition function, are deferred to the appendix, which hinders understanding. Furthermore, the figures, especially those in Section 5, need to be polished for better readability.*
>
> Thank you for pointing this out. We have updated the writing to have more of the key components in the main text, along with improved figures.
>
> > *Classification of Related Work: A minor point—another reference by Malkomes et al. (2021) is arguably classified as a Level Set Estimation (LSE) method. From my perspective, it also addresses identifying sparse, diversified coverage of constrained optimal candidates.*
>
> Thank you for pointing that out. We agree with you that Malkomes et al. (2021) aims to identify a diverse set of points within a hard-constrained region. Malkomes et al. (2021) terms it as “constraint active search”, specifically, which we have updated in the writing.
>
> > *A follow-up work by Zhang and Golovin (2020) builds on Paria et al. (2019) and could allow for a more consistent analysis in terms of optimization performance, rather than assuming Algorithm 1 offers sufficiently dense coverage. The authors might consider incorporating methods from Zhang and Golovin (2020) to improve the completeness of their analysis.*
>
> Thank you for bringing up that suggestion.
> - **Based on your suggestion, we added into the writing a theorem which shows a lower bound for the SHF utility ratio as $t \rightarrow \infty$**. Overall, we show that as $t \rightarrow \infty$, the SHF utility ratio approaches 1 (Section A.3 of the paper).
> - This intuitively proves that our proposed MoSH-Sparse is able to offer sufficiently dense coverage, as intended.
> - To prove the theorem, we leveraged the Lipschitz continuity of our proposed soft-hard bound utility function as well as a regret decomposition form from Russo and Van Roy (2014) and Paria et al. (2020).
> - The proof is in the Appendix and referenced in the main text. Thank you again for this suggestion as we believe it greatly improves our paper.
>
> Again, thank you very much for your feedback. We hope we've addressed your concerns. Please let us know if there are any additional points to discuss.

---

> ### Author Response · Authors · 2024-11-23
> **Author Follow-Up**
>
> Dear Reviewer mTMT, since the discussion period is ending in a few days (Nov. 26), we would love to hear if our response has addressed your concerns. Thank you and please let us know if you have any other questions!

---

### Author Response · Authors · 2024-11-21
**High-Level Comment From Authors**

Thank you to all of the reviewers for your thoughtful feedback! We’ve worked to address all of your comments and questions in the individual threads and wanted to highlight two high-level points to all of the reviewers.

1. **Novelty Of Our Proposed Method**

**Our primary contribution is the novel conceptual framework of soft-hard bounds for multi-objective optimization (formalization of the problem, our proposed method, soft-hard evaluation metrics, and broad set of experiments (across cancer treatment, LLM personalization, deep learning model selection, and more), all native to this novel setting).** To the best of our knowledge, this setting has not been studied before in the multi-objective optimization setting, yet is ubiquitous in many real-world problems. As a result, we are the first to formalize this problem with a simple two-step process which returns an easily interpretable, small set of points to the user according to the soft-hard bounds they had specified. As part of this novel conceptual framework, we also introduced novel evaluation metrics which are native to this soft-hard setting – rather than reusing existing ones. We additionally rigorously evaluated our setting, both theoretically and empirically, while also introducing a novel experiment on multi-objective personalization for LLMs (which had previously not been evaluated on in the multi-objective optimization literature, but shows our setting’s broad relevance).

2. **Additional Theoretical And Empirical Analyses/Experiments**

In response to the valuable feedback, we have added the following three key changes:
- **Theoretical analyses for step 1 of our proposed method, MoSH-Dense, which guarantees its behavior** (as suggested by Reviewers mTMT, vZPB, and qibo) (Appendix A.3 of the paper)
- **Empirical results demonstrating the impact of our max-min to max-average conversion for MoSH-Dense** (as suggested by Reviewer qibo) (Appendix A.1.1 of the paper)
- **Experimental results for the novel multi-objective large language model (LLM) personalization setting** (to showcase our method’s broad applicability and novelty, as mentioned by Reviewer wJhw) (Section 5.3.2 of the paper). For the LLM experiment, we use soft-hard bounds on two competing objectives - conciseness and informativeness - and demonstrate our method's superior ability in consistently sampling output responses within such soft and hard bounds.

---

> ### Author Response · Authors · 2024-12-04
> **Follow-Up High-Level Comment From Authors**
>
> In addition to the above, we believe we have addressed all of the remaining reviewers' comments in the individual threads. The high-level changes are highlighted below:
>
> - **Clarification of a couple related works** (as mentioned by Reviewer mTMT). Clarified Malkomes et al. (2021) as a work on "constraint active search" and updated description of Zuluaga et al. (2016).
> - **Improved readability of paper** (as mentioned by Reviewer mTMT). Brought the Greedy Submodular Partial Cover (GPC) algorithm from the Appendix into the main text and updated readability of all plots/figures.
> - **Clarified usage of SHF Utility Ratios** (as mentioned by Reviewer mTMT). Highlighted their intuitive interpretation and usage as an evaluation metric in Section 2.3.
> - **Emphasized the theoretical guarantees for both steps of our two-step algorithm** (as mentioned by Reviewer vZPB). Theoretical guarantee for step 1 added during discussion stage.
> - **Clarified details around our proposed Soft-Hard Functions (SHFs)** (as suggested by Reviewers qibo and wJhw). Improved motivation for the piecewise-linear utility form and better motivated the saturation threshold in Section 2.2.
>
> Again, we thank all of the reviewers for your valuable thoughts. We believe that all your feedback has certainly strengthened our paper, and we sincerely hope that our added comments and changes address your questions. We are very excited about the introduction of our soft-hard bounds setting and believe it will pave the way for countless other research and practical applications.

---

### Author Response · Authors · 2024-11-29
**Official Comment From Authors**

We sincerely thank all the reviewers for their time and effort in evaluating our work. As the updated discussion deadline approaches (Dec. 2), we would appreciate any feedback on our rebuttal, especially if there are remaining concerns that have not yet been addressed. Thank you!

---

> ### Comment · Reviewer_vZPb · 2024-12-01
> **Thank you**
>
> Dear authors,
>
> the discussion tends to be lively, if at least one of the reviewers is willing to champion your paper. If, as it is unfortunately the case here, all reviewers agree that the paper is not good enough to be accepted, it may be harder to have a lively discussion.

---

### Meta-Review · Area_Chair_ziSB · 2024-12-21

**Metareview:**

The paper studies multi-objective optimization, with an emphasis on incorporating soft and hard constraints into the optimization framework. The paper introduces a two-stage approach that first generates a set of potential solutions and it then refines this set by leveraging algorithms for robust submodular optimization.

The reviewers appreciated the importance of the problem studied for practical applications. The proposed approach via submodular optimization provides a sound approach for a problem that is intractable in general. The reviewers raised significant concerns regarding  the strength and novelty of the contribution and the clarity of the exposition, as detailed in the reviews. Overall, the consensus among the reviewers was that this work does not meet the high bar for acceptance.

**Additional Comments On Reviewer Discussion:**

The reviewers raised the concerns that the main contribution is not sufficiently strong or novel, and that the exposition lacks clarity. The authors provided detailed responses and revised the manuscript based on the reviewers' feedback. Following the discussion, the reviewers maintained their evaluation that the paper does not meet the threshold for acceptance.

---

### Decision · Program_Chairs · 2025-01-22

Reject